# Budget-Constrained Step-Level Diffusion Caching

**Mingkun Lei**[1]  **Tong Zhao**[1]  **Liangyu Yuan**[1]  **Chi Zhang**[1]

`https://github.com/Westlake-AGI-Lab/BudCache`

## Abstract

Step-level caching accelerates diffusion models by exploiting temporal redundancy across denoising steps. Existing methods make per-step cache decisions using threshold-based heuristics, without directly optimizing for final output quality. As a result, their inference latency varies across inputs and is difficult to control at deployment. In this work, we propose BudCache, which inverts this formulation: rather than letting per-step error thresholds dictate the runtime cost, we fix the compute budget in advance and search for the cache policy that best preserves the final output. To tackle the combinatorial complexity of step selection, we combine Simulated Annealing with deterministic Hill Climbing. This offline search identifies high-quality cache policies within minutes and introduces no online search or thresholding overhead during inference. When the compute budget is very tight, we further introduce cache-aware schedule alignment, which adapts the time discretization to the selected cache policy to reduce cache-induced trajectory mismatch. Experiments on FLUX.1-dev and Wan2.1 show that BudCache achieves better generation quality than heuristic caching baselines under the same inference budgets.

## 1. Introduction

Flow-matching (Lipman et al., 2022; Liu et al., 2022) and diffusion models (Ho et al., 2020; Song et al., 2020) have become standard tools for high-fidelity generative tasks, achieving state-of-the-art results in high-resolution image generation (Podell et al., 2023; Esser et al., 2024; Labs, 2024), video synthesis (Wan et al., 2025; Kong et al., 2024; Team et al., 2025), 3D asset creation (Xiang et al., 2025),

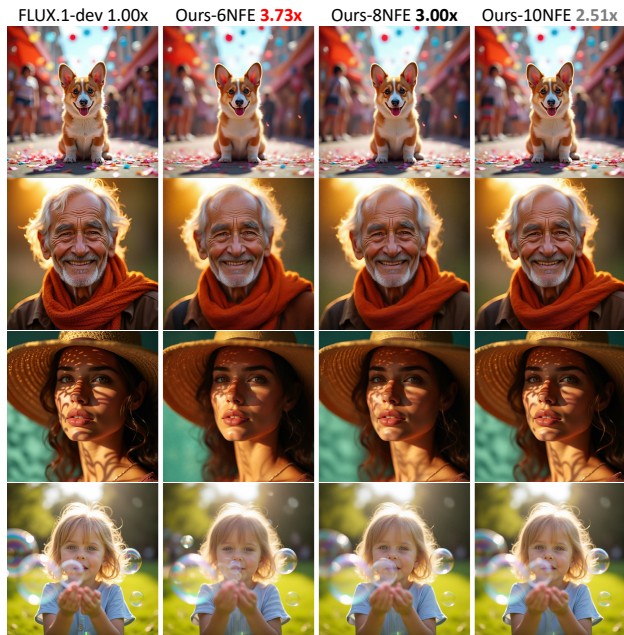

*Figure 1.* **Visualizations of 1024x1024 images generated by FLUX.1-dev** (Labs, 2024). `BudCache` enables aggressive acceleration, achieving up to 3.73x speedup with only 6 NFE while preserving high-frequency details and semantic integrity.

and scientific applications such as protein design (Gruver et al., 2023). Despite their generative capabilities, a central practical bottleneck remains the high computational cost of iterative sampling. Generating a single sample typically requires solving a Probability Flow ODE (PF-ODE), a process that necessitates tens to hundreds of function evaluations (NFE). Since each NFE corresponds to a full forward pass of a large denoising or velocity network, the inference latency can be prohibitive for real-time applications. Consequently, reducing NFE without sacrificing perceptual quality has become a primary objective in accelerating modern diffusion and flow models.

To address this efficiency challenge, step-level caching (Liu et al., 2025a; Ma et al., 2025; Gao et al., 2025; Peng et al., 2025; Bu et al., 2025) has emerged as an appealing acceleration mechanism. Unlike distillation methods (Yin et al., 2024b;a; Ge et al., 2025; Cheng et al., 2025; Lin et al., 2024)

[1]AGI Lab, Westlake University. Correspondence to: Chi Zhang <chizhang@westlake.edu.cn>.

*Proceedings of the $43^{rd}$ International Conference on Machine Learning*, Seoul, South Korea. PMLR 306, 2026. Copyright 2026 by the author(s).

that require expensive retraining, caching strategies reduce NFE by exploiting the temporal redundancy inherent in the generation process. Given that the feature maps and outputs of the diffusion model often evolve slowly between adjacent time steps, a caching method evaluates the full network only at a subset of key steps. For the intermediate steps, it reuses previously cached outputs or internal feature activations to approximate the network prediction. This design choice offers a significant advantage: it serves as a training-free, plug-and-play solution that preserves the weights of the pre-trained model while keeping the sampling process lightweight and making caching easy to apply across different diffusion backbones, including image and video generation models. From this perspective, step-level caching can be viewed as a resource-allocation problem: under a limited computation budget, one must decide which denoising steps deserve fresh model evaluations and which steps can safely reuse cached information.

However, existing caching strategies (Liu et al., 2025a; Ma et al., 2025) do not solve this allocation problem directly. They typically rely on heuristic, threshold-based metrics to decide during inference whether each step should compute or reuse cached information. This design creates two practical issues. First, because cache decisions are triggered by runtime signals, the final number of model evaluations can vary across inputs, making latency difficult to control under a fixed deployment budget. Second, these per-step decisions are not directly optimized for the final generated output, so the resulting cache policy can allocate computation suboptimally for a given NFE budget. These limitations motivate a budget-first formulation: instead of allowing thresholds to determine the runtime cost, we fix the compute budget in advance and search for the cache policy offline.

Beyond deciding which steps to compute or reuse, the selected cache policy also interacts with the solver time discretization. Standard diffusion solvers (Lu et al., 2022; Karras et al., 2022) use pre-defined time schedules that are designed for full model evaluation at every step. When aggressive caching is applied, the sampler reuses stale predictions or activations at many steps, changing the approximation error accumulated along the trajectory. Keeping the original schedule fixed may therefore become suboptimal, especially under very small NFE budgets. This motivates cache-aware schedule alignment, where the time discretization is adapted to the selected cache policy to reduce trajectory mismatch.

Motivated by these observations, we propose BudCache, a budget-constrained framework for step-level diffusion caching. Rather than letting per-step error thresholds determine the runtime cost, BudCache fixes the compute budget in advance and searches for a static cache policy that best preserves the final output. To solve the resulting combinatorial step-selection problem, we combine Simulated

Annealing (Kirkpatrick et al., 1983; Aarts & Korst, 1987) with deterministic Hill Climbing (Aarts & Lenstra, 2018). This offline search identifies high-quality cache policies within minutes using a small calibration set, and requires no online search or thresholding during inference. To further reduce trajectory mismatch under very tight compute budgets, we introduce cache-aware schedule alignment, which adapts the time discretization to the selected cache policy without increasing inference-time NFE. Our contributions are summarized as follows:

- **Budget-constrained caching formulation.** We formulate step-level caching as a discrete cache-policy search problem under a fixed NFE budget. The resulting static policy provides deterministic inference cost and avoids input-dependent threshold triggering.

- **Efficient offline cache-policy search.** We combine Simulated Annealing (Kirkpatrick et al., 1983; Aarts & Korst, 1987) with deterministic Hill Climbing (Aarts & Lenstra, 2018) to search high-quality cache policies over the combinatorial step-selection space.

- **Cache-aware schedule alignment.** When caching is aggressive, we adapt the time discretization to the selected cache policy, reducing cache-induced trajectory mismatch without increasing inference-time NFE.

- **Evaluation across image and video models.** Experiments on FLUX.1-dev (Labs, 2024) and Wan2.1 (Wan et al., 2025) show that BudCache improves generation quality over heuristic caching baselines under matched inference budgets, with additional results across solvers, resolutions, guidance scales, few-step models, and large-scale video models.

## 2. Related Work

**Flow matching models.** Diffusion and flow matching models (Ho et al., 2020; Song et al., 2020; Lipman et al., 2022; Liu et al., 2022) have established themselves as the cornerstone of modern image generation. Representative large-scale text-to-image models (Podell et al., 2023; Esser et al., 2024; Labs, 2024; Labs et al., 2025) scale transformer architectures to billions of parameters to achieve superior fidelity. These models have also become common backbones for downstream applications (Ye et al., 2023; Guo et al., 2024; Lei et al., 2025; Feng et al., 2026; Yuan et al., 2026a). However, this scaling incurs prohibitive computational costs. To mitigate inference latency, the community has developed diverse acceleration strategies, ranging from knowledge distillation (Lin et al., 2024; Yin et al., 2024b;a; Ge et al., 2025; Cheng et al., 2025) and advanced ODE solvers (Lu et al., 2022; Zhu et al., 2025; Wang et al., 2025; Zhao et al., 2026) to training-free caching mechanisms.

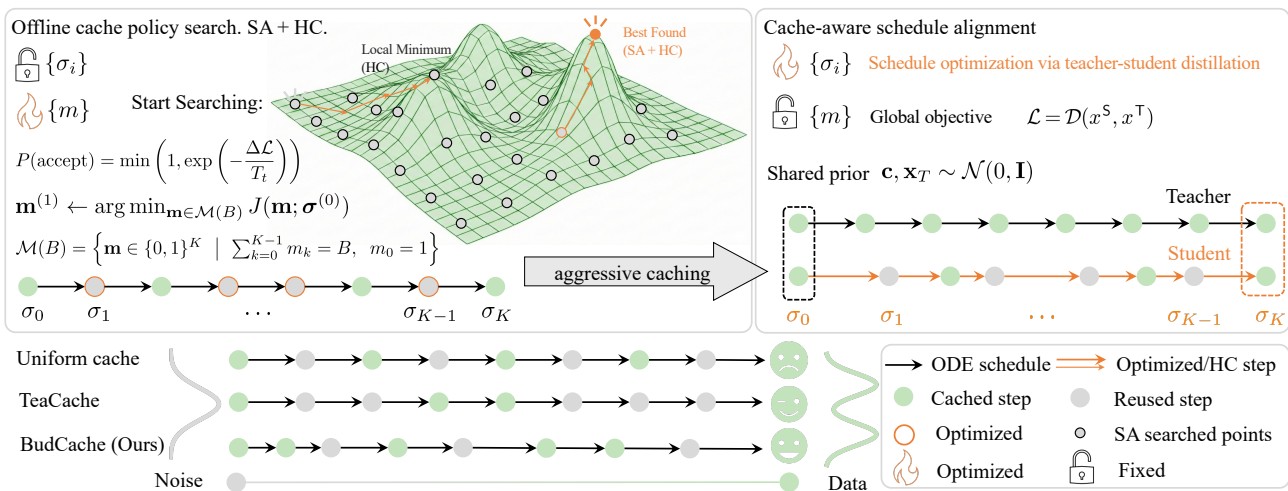

*Figure 2.* **Overview of** `BudCache`. We first perform offline cache-policy search under a fixed NFE budget, using simulated annealing for global exploration and hill climbing for local refinement. When caching is aggressive, we further refine the time discretization $\{\sigma_i\}$ via teacher-student distillation. The bottom panel illustrates the resource allocation produced by `BudCache` compared to heuristic baselines.

**Caching strategies for acceleration.** Exploiting temporal redundancy is key to efficient inference. Fine-grained caching targets redundancy at the layer or token level: DeepCache (Ma et al., 2024), TGATE (Liu et al., 2025b), and BlockDance (Zhang et al., 2025) skip blocks, while methods like ToCa (Zou et al., 2024a) and DuCa (Zou et al., 2024b) prune or merge redundant tokens within attention layers. Recently, step-level caching has emerged as a coarser, often model-agnostic granularity. Current approaches employ diverse selection mechanisms: TeaCache (Liu et al., 2025a) and MagCache (Ma et al., 2025) rely on heuristic thresholds of output differences to dynamically trigger updates. LeMiCa (Gao et al., 2025) advances this by employing dynamic programming to solve for a global caching policy based on error estimation. TaylorSeer (Liu et al., 2025c) takes a different route, synthesizing predictions by combining multi-step cached features in a Taylor-expansion-like manner. While methods like DiCache (Bu et al., 2025) and ERTACache (Peng et al., 2025) further explore error metrics, they still largely follow the dominant threshold-based paradigm. In contrast, we formulate caching as a *budget-constrained optimization problem*, employing global cache strategy search to ensure deterministic latency while maximizing global generation quality.

**Time discretization and schedule optimization.** Standard solvers typically rely on heuristic time schedules (*e.g.*, EDM (Karras et al., 2022)). Recent research has focused on optimizing these schedules to improve sampling efficiency. For instance, Align-Your-Steps (AYS) (Sabour et al., 2024) searches for optimal steps via stochastic algorithms, and LD3 (Tong et al., 2024; Yuan et al., 2026b) learns discretization via differentiable distillation. Our work leverages these advancements but with a distinct motivation: rather than aiming solely for better discretization, we integrate schedule optimization as a compensatory mechanism to correct the trajectory drift induced by aggressive caching.

## 3. Method

### 3.1. Preliminaries

We consider a pre-trained flow-matching model defining a time-dependent vector field $\mathbf{v}_\theta(\mathbf{x}, t, c)$. The generation process is governed by a Probability Flow ODE. Standard numerical solvers approximate the continuous trajectory by discretizing the time horizon $[1, 0]$ into a schedule $\boldsymbol{\sigma} = \{\sigma_0, \ldots, \sigma_K\}$. For the Euler solver, the update rule is:

$$\mathbf{x}_{k+1} \leftarrow \mathbf{x}_k + (\sigma_{k+1} - \sigma_k) \cdot \mathbf{v}_\theta(\mathbf{x}_k, \sigma_k, c). \quad (1)$$

In this formulation, the solver resolution is tightly coupled with the inference cost: the number of logical steps $K$ is identical to the Number of Function Evaluations (NFE).

**Step-level caching.** Recent studies such as TeaCache (Liu et al., 2025a), MagCache (Ma et al., 2025) have demonstrated that the vector field $v_\theta$ exhibits significant temporal redundancy. Consequently, rather than evaluating the model at every logical step, one can accelerate sampling by selectively reusing outputs from preceding steps, thereby significantly reducing the NFE.

### 3.2. Problem formulation

Current caching strategies predominantly rely on heuristic thresholding to trigger reuse. While effective, this paradigm suffers from unpredictable latency and myopic decision-

**Algorithm 1** Global exploration (Simulated Annealing). *Swap* exchanges one cached step with one computed step; *Shift* moves one computed step to an adjacent index. Both preserve $\sum \mathbf{m} = B$.

---

1: **Input:** Budget $B$, $T_0$, $T_{\min}$, $\gamma \in (0, 1)$.
2: **Output:** Initialization $\mathbf{m}_{\text{SA}}$ for Hill Climbing.
3: Initialize $\mathbf{m}$ with uniform spacing, $\sum \mathbf{m} = B$.
4: $E \leftarrow \mathcal{L}(\mathbf{m})$.
5: $\mathbf{m}_{\text{best}} \leftarrow \mathbf{m}$; $E_{\text{best}} \leftarrow E$; $T \leftarrow T_0$.
6: **while** $T > T_{\min}$ **do**
7: $\quad$ $\mathbf{m}' \leftarrow$ Neighbor($\mathbf{m}$) via *Swap* or *Shift*.
8: $\quad$ $E' \leftarrow \mathcal{L}(\mathbf{m}')$; $\Delta E \leftarrow E' - E$.
9: $\quad$ **if** $\Delta E < 0$ **or** rand$(0, 1) < \exp(-\Delta E / T)$ **then**
10: $\quad\quad$ $\mathbf{m} \leftarrow \mathbf{m}'$; $E \leftarrow E'$.
11: $\quad\quad$ **if** $E < E_{\text{best}}$ **then**
12: $\quad\quad\quad$ $\mathbf{m}_{\text{best}} \leftarrow \mathbf{m}$; $E_{\text{best}} \leftarrow E$.
13: $\quad\quad$ **end if**
14: $\quad$ **end if**
15: $\quad$ $T \leftarrow \gamma \cdot T$.
16: **end while**
17: **return** $\mathbf{m}_{\text{SA}} \leftarrow \mathbf{m}_{\text{best}}$.

---

**Algorithm 2** Local refinement (Hill Climbing). $\mathcal{N}(\mathbf{m})$ is the one-step *Swap/Shift* neighborhood of $\mathbf{m}$.

---

1: **Input:** Initialization $\mathbf{m}_{\text{SA}}$ from Algorithm 1.
2: **Output:** Refined cache policy $\mathbf{m}^*$.
3: $\mathbf{m} \leftarrow \mathbf{m}_{\text{SA}}$; $E \leftarrow \mathcal{L}(\mathbf{m})$.
4: **loop**
5: $\quad$ $\mathbf{m}_{\text{new}} \leftarrow \arg\min_{\mathbf{n} \in \mathcal{N}(\mathbf{m})} \mathcal{L}(\mathbf{n})$.
6: $\quad$ $E_{\text{new}} \leftarrow \mathcal{L}(\mathbf{m}_{\text{new}})$.
7: $\quad$ **if** $E_{\text{new}} < E$ **then**
8: $\quad\quad$ $\mathbf{m} \leftarrow \mathbf{m}_{\text{new}}$; $E \leftarrow E_{\text{new}}$.
9: $\quad$ **else**
10: $\quad\quad$ **break**. // Local optimum.
11: $\quad$ **end if**
12: **end loop**
13: **return** $\mathbf{m}^* \leftarrow \mathbf{m}$.

---

making. To address these limitations and strictly control the computational cost, we propose to model the caching decision process via a binary mask $\mathbf{m} \in \{0, 1\}^K$.

We define $m_k = 1$ as a fresh computation step and $m_k = 0$ as a cache reuse step. Let $a(k) = \max\{j \leq k \mid m_j = 1\}$ be the index of the most recent active computation. The cached update rule is formulated as:

$$\mathbf{x}_{k+1} \leftarrow \mathbf{x}_k + (\sigma_{k+1} - \sigma_k) \cdot \mathbf{v}_\theta(\mathbf{x}_{a(k)}, \sigma_{a(k)}, c). \quad (2)$$

This formulation effectively decouples the discretization granularity ($K$) from the inference budget ($B = \sum m_k$).

Based on this formulation, we propose `BudCache`. We invert the standard heuristic paradigm: instead of letting error thresholds passively dictate the cost, we enforce a fixed budget $B$ and optimize final-output fidelity. We formulate this as a joint optimization problem:

$$\min_{\mathbf{m}, \boldsymbol{\sigma}} \quad \mathbb{E}_{x^{(0)}, c} \left[ \mathcal{L}_{\text{distill}} \left( x^{\mathsf{S}}(\mathbf{m}, \boldsymbol{\sigma}), x^{\mathsf{T}} \right) \right], \quad (3)$$

$$\text{s.t.} \quad \sum_{k=0}^{K-1} m_k = B, \quad m_0 = 1.$$

Here, optimizing the mask $\mathbf{m}$ selects the most information-dense steps, while optimizing the schedule $\boldsymbol{\sigma}$ can further adapt the step sizes to mitigate trajectory drift induced by caching errors.

### 3.3. Cache policy search via hybrid optimization

Our objective is to identify the caching policy $\mathbf{m}$ that maximizes generation fidelity under a strict budget constraint

$B$. This presents a formidable combinatorial challenge: the search space is discrete ($\binom{K}{B}$ possibilities), and the optimization landscape is highly non-convex. Due to the sequential nature of diffusion sampling, a caching decision at an early step alters the trajectory drift, cascading errors to all subsequent steps.

Standard greedy approaches, such as simple Hill Climbing (Aarts & Lenstra, 2018), are ill-suited for this task. Because they rely on myopic local moves, they can stagnate in local optima, configurations that are locally stable but globally sub-optimal. While repeating the greedy search from multiple random initializations could theoretically mitigate this, it is computationally prohibitive given the vastness of the search space.

To robustly navigate this landscape, we employ a Hybrid Optimization Strategy detailed in Algorithms 1 and 2 that effectively decouples the search into two distinct phases: probabilistic exploration via Simulated Annealing (SA) (Kirkpatrick et al., 1983; Aarts & Korst, 1987) and deterministic refinement via Hill Climbing (HC) (Aarts & Lenstra, 2018), as shown in Figure 2.

**Phase 1: global exploration (SA).** We first employ Simulated Annealing to identify a broad basin of high-quality solutions. We formulate the search as an energy minimization process over the budget-constrained manifold $\mathcal{M}_B = \{\mathbf{m} \in \{0, 1\}^K \mid \sum m_k = B\}$. To explore $\mathcal{M}_B$ while strictly preserving the budget, we define the transition dynamics using two topological operators: (1) **Swap**: Exchange a compute bit (1) with a reuse bit (0) at random positions. (2) **Shift**: Displace a compute bit to an adjacent available position. At each iteration $t$, we generate a candidate $\mathbf{m}'$ by applying a random operator. The candidate is accepted based on the Metropolis criterion:

$$P(\text{accept}) = \min\left(1, \exp\left(-\frac{\Delta\mathcal{L}}{T_t}\right)\right). \quad (4)$$

---

**Algorithm 3** Cache-aware schedule alignment

---

1: **Input:** Frozen network $v_\theta$, cache policy $\mathbf{m}^*$, reference schedule $\sigma_{\text{ref}}$, prompt distribution $\mathcal{P}$, learning rate $\eta$.
2: **Output:** Optimized Schedule $\sigma^*$.
3: Initialize learnable schedule $\sigma$ from $\sigma_{\text{ref}}$.
4: // Optimization Loop
5: **while** not converged **do**
6:     Sample noise $x^{(0)} \sim \mathcal{N}(0, I)$ and prompts $c \sim \mathcal{P}$.
7:     // 1. Teacher Generation (Full-Compute Reference)
8:     Generate teacher trajectory $x^{\mathsf{T}}$ using $v_\theta$ with $\sigma_{\text{ref}}$ (full computation).
9:     // 2. Student Generation (Cache-Aware)
10:    Generate student trajectory $x^{\mathsf{S}}$ using $v_\theta$ with current $\sigma$ and caching policy $\mathbf{m}^*$ (Cached Solver).
11:    // 3. Update Schedule
12:    Compute Loss $\mathcal{L} = \mathcal{D}(\mathbf{x}^{\mathsf{S}}, \mathbf{x}^{\mathsf{T}})$.
13:    Update $\sigma \leftarrow \sigma - \eta \nabla_\sigma \mathcal{L}$.
14: **end while**
15: **return** $\sigma^* \leftarrow \sigma$.

---

The high initial temperature $T_t$ allows the system to accept "energetically unfavorable" moves (*i.e.*, higher loss), enabling it to traverse barriers and escape the local optima that would trap a pure greedy solver.

**Phase 2: deterministic refinement (HC).** As the temperature $T_t \to 0$, the stochastic exploration ceases, and we transition to a deterministic greedy polish phase. This phase ensures that the final solution $\mathbf{m}^*$ converges to the precise bottom of the current valley. Unlike the stochastic neighbor selection in SA, we define the neighborhood $\mathcal{N}(\mathbf{m})$ as the set of *all* masks reachable by a single Swap or Shift operation. In each step, we rigorously evaluate all candidates in $\mathcal{N}(\mathbf{m})$ and greedily transition to the neighbor yielding the maximum loss reduction. This process repeats until no further gain is possible, efficiently locking in the locally optimal configuration.

The derived mask $\mathbf{m}^*$ represents a high-quality resource allocation strategy. Unlike myopic heuristics that yield unpredictable inference costs due to data-dependent triggering, our approach guarantees strict latency adherence, *i.e.*, NFE $= B$, and maximizes visual fidelity via a low-cost calibration. Crucially, since the search is performed offline, it incurs zero overhead during deployment.

However, while $\mathbf{m}^*$ successfully identifies the most information-dense steps, applying a sparse mask to the original solver inevitably alters the trajectory dynamics and error profile. We address this induced drift by recalibrating the time schedule in the subsequent section.

### 3.4. Cache-aware schedule alignment

While the optimized mask $\mathbf{m}^*$ determines which steps use

fresh computation, it is still applied on top of a time schedule that is originally designed for full model evaluation at every step. Under aggressive caching, many updates use stale predictions or activations, which changes the approximation error accumulated along the sampling trajectory. This effect becomes more pronounced under very tight compute budgets, where each step carries a larger portion of the total update. To further improve fidelity, we refine the time discretization after the cache mask is fixed.

**Motivation.** Consider the Euler update at step $k$. The cached solver utilizes a stale vector $\mathbf{v}(\mathbf{x}_{a(k)})$ instead of the precise $\mathbf{v}(\mathbf{x}_k)$, introducing a local approximation error $\mathbf{e}_k = \mathbf{v}(\mathbf{x}_{a(k)}) - \mathbf{v}(\mathbf{x}_k)$. The actual update term becomes:

$$\Delta\mathbf{x}_k = \Delta\sigma_k \cdot (\mathbf{v}(\mathbf{x}_k) + \mathbf{e}_k) = \underbrace{\Delta\sigma_k\mathbf{v}(\mathbf{x}_k)}_{\text{Ideal Update}} + \underbrace{\Delta\sigma_k\mathbf{e}_k}_{\text{Drift Term}}. \quad (5)$$

Crucially, the step size $\Delta\sigma_k$ acts as a multiplicative gain on the error $\mathbf{e}_k$. If step sizes remain uniform, large errors (typically in high-curvature regions) are heavily weighted, severely degrading the trajectory. To minimize cumulative deviation, the solver must adaptively *shrink* $\Delta\sigma_k$ where caching introduces significant variance (large $\|\mathbf{e}_k\|$), and *expand* it where the approximation is accurate.

**Schedule alignment via output matching.** Directly estimating the local cache error $\mathbf{e}_k$ for every candidate schedule is impractical, so we optimize the schedule through the final output. Given the fixed cache mask $\mathbf{m}^*$, we treat the schedule $\sigma = \{\sigma_i\}$ as learnable parameters and optimize it to match the output of the full solver. Let $\mathbf{x}^{\mathsf{S}}(\mathbf{m}^*, \sigma)$ denote the final output of the cached sampler using mask $\mathbf{m}^*$ and schedule $\sigma$, and let $\mathbf{x}^{\mathsf{T}}$ denote the final output of the full solver. We optimize

$$\sigma^* = \arg\min_\sigma \mathbb{E}\left[\mathcal{D}\left(\mathbf{x}^{\mathsf{S}}(\mathbf{m}^*, \sigma), \mathbf{x}^{\mathsf{T}}\right)\right], \quad (6)$$

where $\mathcal{D}(\cdot, \cdot)$ is an output-level discrepancy measure, such as MSE. This objective aligns the cached sampler with the full solver at the output level. Since the schedule controls the step sizes along the trajectory, optimizing $\sigma$ allows the sampler to redistribute temporal resolution according to the error pattern induced by the fixed cache policy. After optimization, the learned schedule is fixed and used together with the searched cache mask during inference, adding no extra model evaluations.

## 4. Experiments

### 4.1. Experimental setup

**Metrics and baselines.** We follow the default settings of each baseline for fair comparison, and evaluate performance from two aspects: efficiency and visual quality. Efficiency is measured by latency and relative speedup. For image

| Original | ERTACache | TaylorSeer | MagCache | DiCache | LeMiCa | TeaCache | Ours-ultra | Ours-fast | Ours-base |
| 1.00x | 3.00x | 2.82x | 2.74x | 2.66x | 2.51x | 2.51x | 3.00x | 2.74x | 2.51x |

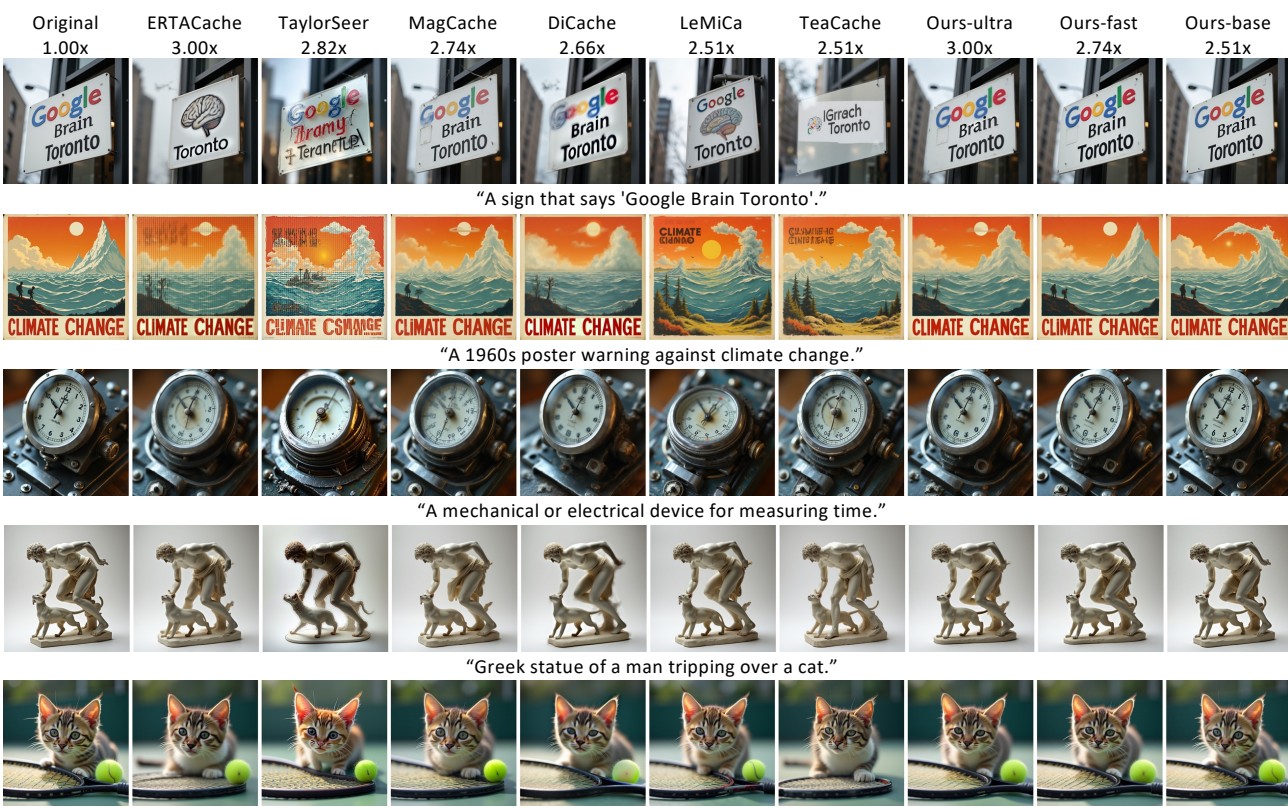

"A sign that says 'Google Brain Toronto'."

"A 1960s poster warning against climate change."

"A mechanical or electrical device for measuring time."

"Greek statue of a man tripping over a cat."

"A cat on the right of a tennis racket."

*Figure 3.* **Qualitative comparison with baselines on FLUX.1-dev** (Labs, 2024). `BudCache` better preserves semantic consistency and fine details, especially for challenging cases such as text rendering and complex geometric structures. Best viewed zoomed in.

generation, visual quality is evaluated using reference-based metrics, including LPIPS (Zhang et al., 2018), SSIM (Wang & Bovik, 2002), and PSNR, as well as reference-free metrics, including CLIP-Score (Radford et al., 2021), ImageReward (Xu et al., 2023), and HPSv2.1 (Wu et al., 2023). We benchmark against representative diffusion caching methods, including TeaCache (Liu et al., 2025a), MagCache (Ma et al., 2025), LeMiCa (Gao et al., 2025), DiCache (Bu et al., 2025), ERTACache (Peng et al., 2025) and TaylorSeer (Liu et al., 2025c), using their official implementations. Image experiments are conducted on DrawBench (Saharia et al., 2022), where we generate 4 samples per prompt with different random seeds and report averaged results. For video generation, we evaluate on Wan2.1-T2V-1.3B (Wan et al., 2025) and compare with TeaCache (Liu et al., 2025a) on a carefully curated set of 100 prompts. We report latency for efficiency and PSNR, SSIM (Wang & Bovik, 2002), and LPIPS (Zhang et al., 2018) for reconstruction fidelity.

**Implementation details.** Unless otherwise specified, all experiments are conducted on NVIDIA H100 GPUs using the PyTorch framework. For comprehensive algorithmic implementation details and a theoretical efficiency analysis,

please refer to Section A.

### 4.2. Comparison with State-of-the-Art methods

**Quantitative analysis.** Table 1 presents the performance comparison on the DrawBench benchmark (Saharia et al., 2022), where `BudCache` demonstrates comprehensive superiority. When constrained to identical latency budgets, our method consistently delivers higher visual fidelity. For instance, at the 2.51x speedup tier, `BudCache-base` achieves an LPIPS (Zhang et al., 2018) score of **0.1759**, significantly outperforming the previous state-of-the-art TeaCache (Liu et al., 2025a) and LeMiCa (Gao et al., 2025). Similarly, in the high-speed regime of 3.00x, `BudCache-ultra` surpasses ERTACache (Peng et al., 2025) by a substantial margin in both reconstruction error and human preference metrics. Beyond direct budget comparisons, our caching policy is efficient enough to outperform baselines that consume significantly more computational resources. As evidenced by the results, our aggressive `BudCache-ultra` variant at 3.00x speedup yields better perceptual quality compared to TeaCache (Liu et al., 2025a), despite the latter operating at a much lower speed of 2.51x.

*Table 1.* **Quantitative comparison with baselines**. Our method consistently outperforms state-of-the-art caching baselines across speed and quality. `BudCache` achieves higher fidelity under identical latency budgets, outperforming MagCache (Ma et al., 2025) at 2.74× speedup. It also exceeds baselines that operate at higher computational costs, as `BudCache-ultra` surpasses TeaCache (Liu et al., 2025a), demonstrating the effectiveness of budget-constrained optimization. The best results are in **bold**, the second best are underlined.

| Method | Efficiency | | Visual Quality | | | | | | |
|---|---|---|---|---|---|---|---|---|---|
| | **Latency (s)↓** | **Speedup↑** | **LPIPS↓** | **SSIM↑** | **PSNR↑** | **ImageReward↑** | **CLIP(B)↑** | **CLIP(L)↑** | **HPSv2.1↑** |
| Original: 28 steps | 6.90 | 1.00x | – | – | – | 1.0138 | 32.300 | 27.581 | 0.2980 |
| ERTACache (Peng et al., 2025) | 2.30 | 3.00x | 0.3208 | 0.7603 | 20.343 | 0.9243 | 32.387 | 27.516 | 0.2835 |
| TaylorSeer (Liu et al., 2025c) | 2.45 | 2.82x | 0.4490 | 0.6454 | 16.092 | 0.8857 | 32.164 | 27.264 | 0.2841 |
| MagCache (Ma et al., 2025) | 2.52 | 2.74x | 0.2498 | 0.7854 | 20.893 | 0.9544 | 32.346 | 27.525 | 0.2910 |
| DiCache (Bu et al., 2025) | 2.59 | 2.66x | 0.3286 | 0.7708 | 21.670 | 0.8091 | 32.321 | 27.463 | 0.2707 |
| LeMiCa (Gao et al., 2025) | 2.75 | 2.51x | 0.3138 | 0.7397 | 18.411 | 0.9750 | 32.369 | 27.555 | 0.2933 |
| TeaCache (Liu et al., 2025a) | 2.75 | 2.51x | 0.3218 | 0.7362 | 18.401 | 0.9708 | 32.285 | 27.505 | 0.2904 |
| `BudCache-ultra` (8 NFE) | 2.30 | 3.00x | 0.2479 | 0.7917 | 22.163 | 0.9495 | 32.392 | 27.474 | 0.2884 |
| `BudCache-fast` (9 NFE) | 2.52 | 2.74x | 0.2020 | 0.8201 | 23.586 | 0.9633 | 32.422 | **27.598** | 0.2921 |
| `BudCache-base` (10 NFE) | 2.75 | 2.51x | **0.1759** | **0.8374** | **24.903** | **0.9782** | **32.446** | 27.584 | **0.2940** |

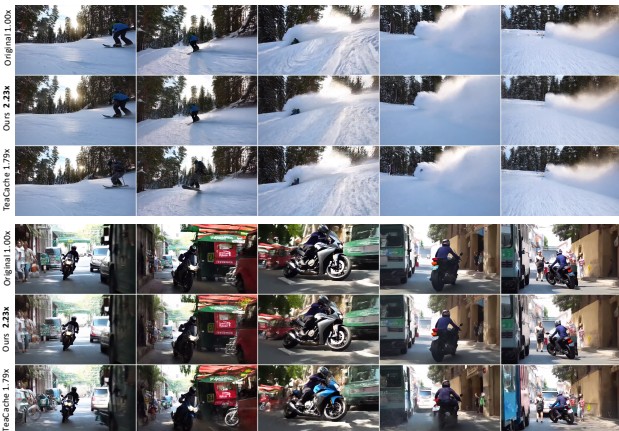

*Figure 4.* **Visualizations of videos generated by Wan2.1-T2V-1.3B (Wan et al., 2025).** We generate videos at a resolution of 832 × 480 with 81 frames, and uniformly sample 5 frames for visualization. `BudCache` achieves stronger acceleration while preserving better visual quality, including finer details and more consistent semantics across frames.

*Table 2.* **Quantitative comparison on Wan2.1-T2V-1.3B (Wan et al., 2025).** We evaluate on a carefully curated set of 100 prompts using an NVIDIA A800 GPU. `BudCache` achieves the lowest latency while consistently improving reconstruction fidelity over TeaCache (Liu et al., 2025a) in terms of PSNR, SSIM (Wang & Bovik, 2002), and LPIPS (Zhang et al., 2018).

| Method | Latency (s)↓ | PSNR↑ | SSIM↑ | LPIPS↓ |
|---|---|---|---|---|
| Full | 189s | – | – | – |
| TeaCache | 100s | 21.52 | 0.7494 | 0.1929 |
| `BudCache` | **82s** | **25.93** | **0.8502** | **0.1167** |

This confirms that our optimized caching strategy utilizes the NFE budget far more effectively than heuristic thresholding. We further validate the effectiveness of `BudCache` on video generation with Wan2.1-T2V-1.3B (Wan et al., 2025). As shown in Table 2, `BudCache` achieves better reconstruction fidelity than TeaCache while using lower latency, demonstrating that the proposed caching strategy generalizes beyond image generation and remains effective for video diffusion models.

**Qualitative analysis.** Figure 3 provides a visual comparison of samples generated by FLUX.1-dev (Labs, 2024) across different caching strategies. The results highlight the distinct advantage of `BudCache` in maintaining both semantic alignment and fine-grained fidelity. In the first two rows, which focus on typographic generation, heuristic-based baselines frequently struggle with character consistency and spelling accuracy. For example, TaylorSeer (Liu et al., 2025c) produces garbled text artifacts while Tea-Cache (Liu et al., 2025a) introduces noticeable typos in the signage. In contrast, our approach correctly renders the phrases "Google Brain Toronto" and "CLIMATE CHANGE" with high precision, closely matching the quality of the original full-step results. Moving to complex structural details in the third and fourth rows, our method demonstrates superior capability in synthesizing intricate geometries. The numerals on the mechanical clock face remain sharp and legible with our approach, whereas competing methods tend to blur these minute features. Similarly, in the statue example, `BudCache` accurately preserves the anatomical details and the spatial interaction between the figures, which often appear distorted or smoothed out in other baselines. The final row further confirms that our global caching strategy effectively maintains texture fidelity and correct spatial positioning, yielding results that are visually close to the full-step reference. We also provide qualitative video results on Wan2.1-T2V-1.3B (Wan et al., 2025) in Figure 4. Compared with the baseline, `BudCache` better preserves fine-grained visual attributes under acceleration, such as the color of the character's clothing and the motorcycle, demonstrating stronger fidelity preservation for video generation.

*Table 3.* **Search-transfer generalization across inference settings**. A cache configuration searched in one source setting is directly transferred to other solvers, resolutions, and CFG scales on FLUX.1-dev (Labs, 2024) using DrawBench (Saharia et al., 2022) prompts, demonstrating consistent generalization of the searched configuration across inference settings.

| Setting | Method | Reconstruction Quality | | |
|---|---|---|---|---|
| | | LPIPS↓ | SSIM↑ | PSNR↑ |
| iPNDM(2M) | TeaCache | 0.4124 | 0.6569 | 15.35 |
| | BudCache-base | **0.1632** | **0.8337** | **23.80** |
| DPM-Solver++(2M) | TeaCache | 0.3912 | 0.6717 | 15.97 |
| | BudCache-base | **0.1668** | **0.8333** | **23.80** |
| 512×512 | TeaCache | 0.3154 | 0.6693 | 16.77 |
| | BudCache-base | **0.1167** | **0.8465** | **24.70** |
| 768×1024 | TeaCache | 0.3634 | 0.6969 | 16.91 |
| | BudCache-base | **0.1370** | **0.8663** | **25.47** |
| CFG = 5 | TeaCache | 0.3749 | 0.6878 | 16.49 |
| | BudCache-base | **0.1536** | **0.8560** | **24.75** |
| CFG = 7 | TeaCache | 0.4254 | 0.6656 | 15.60 |
| | BudCache-base | **0.1649** | **0.8380** | **24.42** |

*Table 4.* **Impact of cache-aware schedule alignment in low-NFE regimes.** We compare visual quality metrics under strict low-NFE budgets. "Opt." denotes our lightweight cache-aware schedule alignment. BudCache consistently improves perceptual and semantic quality under similar latency, showing the synergy between step caching and time discretization. The best results under each NFE budget are in **bold**.

| Configuration | Latency (s)↓ | ImageReward↑ | CLIP(L)↑ | HPSv2.1↑ |
|---|---|---|---|---|
| Original (5 NFE) | 1.61 | 0.5896 | 27.184 | 0.2583 |
| Original-Opt. (5 NFE) | 1.61 | 0.5991 | 27.169 | 0.2601 |
| BudCache (5 NFE) | 1.64 | 0.7808 | 27.322 | 0.2657 |
| BudCache-Opt. (5 NFE) | 1.64 | **0.7888** | **27.442** | **0.2682** |
| Original (6 NFE) | 1.83 | 0.7236 | 27.286 | 0.2622 |
| Original-Opt. (6 NFE) | 1.83 | 0.7136 | 27.273 | 0.2632 |
| BudCache (6 NFE) | 1.85 | 0.8221 | **27.370** | 0.2712 |
| BudCache-Opt. (6 NFE) | 1.85 | **0.8289** | 27.355 | **0.2742** |
| Original (7 NFE) | 2.05 | 0.8519 | 27.437 | 0.2801 |
| Original-Opt. (7 NFE) | 2.05 | 0.8099 | 27.469 | 0.2804 |
| BudCache (7 NFE) | 2.06 | 0.8895 | 27.423 | 0.2807 |
| BudCache-Opt. (7 NFE) | 2.06 | **0.9004** | **27.603** | **0.2861** |

Additional visualizations are provided in Section B.

### 4.3. Cache-aware schedule alignment

We further evaluate cache-aware schedule alignment under tight low-NFE budgets of 5, 6, and 7 NFE. We focus on this regime because, after the cache policy has been optimized, further schedule refinement brings limited gains at higher NFE budgets, where the default discretization is already sufficiently effective. Under very small NFE budgets, each step carries a larger portion of the trajectory update, making the sampler more sensitive to cache-induced mismatch. As shown in Table 4, simply reducing the number of sampling steps in the standard solver leads to severe quality degradation. In contrast, BudCache significantly outperforms these uniform low-step baselines, with a 0.19 increase in

*Table 5.* **Generalization to GenEval prompts**. A cache configuration searched on the source setting is directly transferred to GenEval (Ghosh et al., 2023) prompts on FLUX.1-dev (Labs, 2024). BudCache-base consistently improves reconstruction quality over TeaCache under the same inference budget.

| Setting | LPIPS↓ | SSIM↑ | PSNR↑ | HPSv2.1↑ | ImageReward↑ | CLIP(L)↑ |
|---|---|---|---|---|---|---|
| Full | – | – | – | 0.3086 | 1.0046 | 27.84 |
| TeaCache | 0.3349 | 0.7586 | 18.56 | 0.2995 | 0.9393 | **27.85** |
| BudCache-base | **0.1388** | **0.8802** | **26.57** | **0.3055** | **0.9710** | 27.83 |

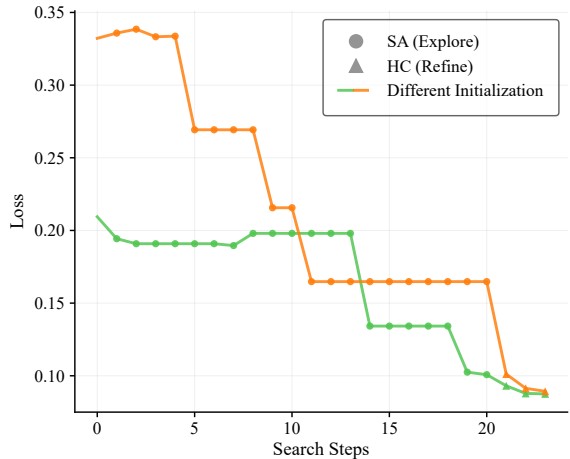

*Figure 5.* **Convergence consistency across random initializations**. We visualize the Latent MSE reduction for two search trials initialized with different random policies. The optimization proceeds through Simulated Annealing shown as circles followed by Greedy Polish shown as triangles. Both trajectories rapidly close the initial performance gap and converge to a nearly identical minimum, demonstrating that the proposed method is robust to initialization randomness and capable of reliably locating high-quality solutions.

ImageReward (Xu et al., 2023) at 5 NFE. This shows that selectively reusing features through an optimized cache policy is more effective than naively reducing denoising steps. We also observe that schedule alignment alone brings limited gains, while combining it with our caching strategy achieves the best overall performance in this low-NFE regime. This suggests that schedule alignment is a useful complementary refinement: once the cache policy is fixed, adapting the time discretization can further reduce the mismatch between the cached sampler and the full solver. Notably, this refinement is efficient; for instance, optimizing the schedule for 7 NFE requires only 6 minutes on 4 NVIDIA H100 GPUs and does not increase inference-time NFE.

### 4.4. Ablation and analysis

**Robustness analysis.** As shown in Figure 5, we run the search from two distinct initial masks with different initial losses. Both trajectories rapidly reduce the calibration objective during the SA exploration phase and further converge after the greedy polish stage. The final losses are nearly identical, indicating that the hybrid search is stable to initial-

*Table 6.* **Compatibility with distilled few-step models**. We evaluate `BudCache` on Hyper-SD (Ren et al., 2024) under the 8-step setting using DrawBench (Saharia et al., 2022) prompts. At the same latency, `BudCache` consistently improves reconstruction and semantic quality over TeaCache (Liu et al., 2025a), showing its compatibility with few-step models.

| Setting | LPIPS↓ | SSIM↑ | PSNR↑ | HPSv2.1↑ | ImageReward↑ | CLIP(L)↑ |
|---|---|---|---|---|---|---|
| Full (8 steps) | – | – | – | 0.3076 | 1.0175 | 27.43 |
| TeaCache | 0.1982 | 0.7631 | 21.56 | 0.3038 | 0.9873 | 27.43 |
| BudCache-base | **0.1417** | **0.8122** | **25.28** | **0.3061** | **1.0096** | **27.58** |

*Table 7.* **Mask transferability across model sizes.** We transfer a mask searched on Wan2.1-T2V-1.3B (Wan et al., 2025) to Wan2.1-T2V-14B (Wan et al., 2025) and evaluate on a curated set of 30 prompts. `BudCache` achieves lower latency and better reconstruction fidelity than TeaCache (Liu et al., 2025a), showing that the searched mask transfers across model sizes.

| Method | Latency (s)↓ | LPIPS↓ | SSIM↑ | PSNR↑ |
|---|---|---|---|---|
| Full | 515s | – | – | – |
| TeaCache | 291s | 0.0639 | 0.8932 | 26.70 |
| BudCache | **223s** | **0.0618** | 0.8932 | **27.94** |

ization and reliably finds high-quality cache policies within a small offline budget.

**Generalization analysis.** As shown in Table 3, we evaluate whether the cache configuration obtained by search is tied to the calibration setting described in Section A or can transfer to different inference conditions. We search the cache configuration once in the calibration setting and directly apply it, without re-searching, to other conditions on FLUX.1-dev (Labs, 2024) using DrawBench (Saharia et al., 2022) prompts. All methods are compared under the same acceleration ratio. Across all tested conditions, `BudCache-base` consistently outperforms TeaCache in LPIPS (Zhang et al., 2018), SSIM (Wang & Bovik, 2002), and PSNR, indicating that the searched configuration is not overfitted to the calibration setting. These results demonstrate that our search discovers a transferable caching policy that generalizes well across inference conditions. Beyond DrawBench (Saharia et al., 2022), we further evaluate the transferability of the searched caching policy on another benchmark, GenEval (Ghosh et al., 2023). As shown in Table 5, the caching policy searched on the source setting is directly transferred to GenEval (Ghosh et al., 2023) prompts on FLUX.1-dev (Labs, 2024) without re-searching.

**Applicability to few-step and large-scale video models.** We first examine `BudCache` on Hyper-SD (Ren et al., 2024), a distilled few-step model with only 8 denoising steps, where we re-search the cache configuration on Hyper-SD itself. As shown in Table 6 and Figure 6, `BudCache` improves both quantitative metrics and qualitative visual fidelity over TeaCache (Liu et al., 2025a) under comparable acceleration settings, showing that effective step-level caching remains possible in few-step regimes. We then

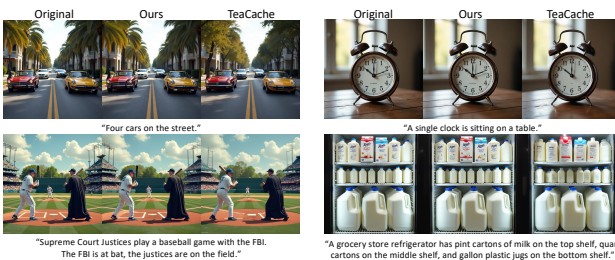

*Figure 6.* **Qualitative comparison on Hyper-SD (Ren et al., 2024).** We compare `BudCache` with TeaCache (Liu et al., 2025a) at the same latency in the distilled few-step setting. `BudCache` better preserves fine-grained details.

study low-cost adaptation for large-scale video generation. Instead of re-searching on the expensive target model, we search the cache mask on Wan2.1-T2V-1.3B (Wan et al., 2025) and directly transfer it to Wan2.1-T2V-14B (Wan et al., 2025). As shown in Table 7, the transferred mask achieves lower latency and better reconstruction fidelity than TeaCache, suggesting that small-model search can provide an economical proxy for accelerating much larger video diffusion models.

## 5. Conclusion

In this work, we addressed two key limitations of heuristic diffusion caching strategies: unpredictable deployment latency and limited alignment with global generation quality. We proposed `BudCache`, a budget-constrained framework that formulates step-level caching as a discrete optimization problem under a fixed inference budget. By combining Simulated Annealing with a deterministic greedy polish, our method efficiently searches the combinatorial cache space and identifies high-quality caching policies that account for global effects across the sampling trajectory. We further showed that schedule optimization complements cache search under tight inference budgets by mitigating cache-induced trajectory mismatch. Extensive experiments across image and video diffusion models demonstrate that `BudCache` achieves better generation quality than existing caching baselines under the same inference budget. These results suggest that budget-aware cache search offers a practical path toward deterministic and latency-efficient diffusion inference.

**Limitations and future work.** The offline search is performed separately for each model and NFE budget, leaving room for more transferable cache policies. Cache-aware schedule alignment could also be further refined, for example by scaling up the calibration set or by jointly optimizing cache policies and time schedules. Another open direction is to better understand the trade-off between global cache policies and instance-adaptive caching for inputs with diverse generation dynamics.

## Impact Statement

This paper presents work whose goal is to advance the field of Machine Learning. There are many potential societal consequences of our work, none of which we feel must be specifically highlighted here.

## Acknowledgements

This work was supported by the National Natural Science Foundation of China (No. 6250070674) and the Zhejiang Leading Innovative and Entrepreneur Team Introduction Program (2024R01007). We thank Jiaxin Yao for helpful discussions.

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

## A. Implementation details

In *Cache policy search via hybrid optimization*, we use the following two prompts for optimization under the calibration setting of Euler sampling, $1024 \times 1024$ resolution, and guidance scale 3.5:

> 1    In a still frame, a stop sign.
>
> 2    a laptop, frozen in time.

**Efficiency of cache policy search.** The computational overhead of identifying the searched caching policy $\mathbf{m}^*$ is minimal. As detailed in Table 8, this search is performed as a one-time offline calibration, imposing no additional latency during inference and requiring no extensive model retraining.

**Impact of search space size.** As shown in Figure 7, we study different total step configurations while fixing the absolute computational budget to 10 NFE. This setting highlights that the relative acceleration ratio alone is not always the most informative criterion: increasing the nominal number of solver steps improves the discretization trajectory, but also requires more aggressive caching under a fixed NFE budget. The resulting quality curve follows an inverted-U shape, suggesting a trade-off between solver discretization error and cache- induced approximation error, with the best balance achieved at $K = 30$.

**Cache-aware schedule alignment details.** The time schedule refinement is performed offline on a lightweight calibration set comprising only 100 prompts randomly sampled from the MS-COCO 2014 validation dataset (Lin et al., 2014). We fix the caching policy $\mathbf{m}^*$ obtained from the search phase and optimize the continuous time steps $\boldsymbol{\sigma}$. The optimization uses the AdamW optimizer with a learning rate of $1 \times 10^{-3}$, a cosine annealing scheduler, and no weight decay. The process is highly efficient, requiring only 1 epoch with a batch size of 1 at a resolution of $1024 \times 1024$. We employ BF16 mixed precision for acceleration. The student model operates under a strict budget such as 5 NFE within a 28 steps logical space, while the teacher utilizes the full 28 steps standard FLUX.1-dev (Labs, 2024; Labs et al., 2025) schedule. The objective is to minimize the MSE loss between the student's and teacher's final generated images.

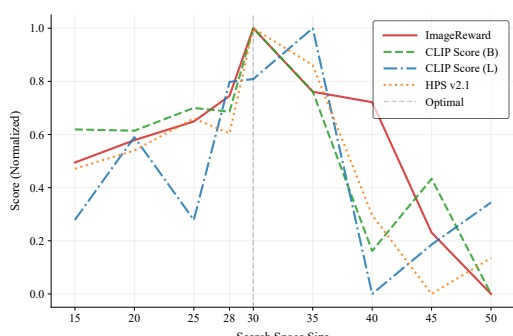

*Figure 7.* **Impact of search space size under a fixed computational budget.** We vary the total solver steps while keeping the budget fixed to 10 NFE.

## B. More results

### B.1. More Qualitative Results

We provide additional qualitative results to further validate the effectiveness of our method. More visual comparisons on FLUX.1-dev (Labs, 2024) are shown in Figures 8 and 9, where `BudCache` consistently preserves semantic alignment and fine-grained details under acceleration. Additional video results on Wan2.1-T2V-1.3B (Wan et al., 2025) are provided in Figure 10, further demonstrating the applicability of our method to video generation.

### B.2. Selected prompts used for visualization

The prompts used to generate the results shown in Figures 1, 4 and 10 are detailed in this section to ensure reproducibility.

> 1    A corgi sitting in a colorful street festival scene, bright balloons and confetti, saturated reds and blues, clear sunny lighting, crisp fur detail, lively atmosphere, cinematic bokeh, ultra sharp.
>
> 2    A smiling elderly man with deep character wrinkles and warm eyes, golden hour sunlight, vivid orange-red scarf, rich skin texture, natural pores, shallow depth of field, soft background blur, documentary portrait style but polished, high dynamic range, vibrant yet natural tones

3    A photorealistic portrait of a woman wearing a wide-brim hat casting patterned shadows across her face, warm sunlight, rich amber and teal color palette, sharp eye detail, fine skin texture, shallow depth of field, cinematic framing, vibrant yet elegant

4    A cheerful portrait of a child blowing soap bubbles in a sunny park, rainbow bubble highlights, saturated greens and blues, sun flare, crisp detail, shallow depth of field, ultra-realistic, vibrant summer mood

5    A snowboarder descends through fresh powder in a pine forest, low follow camera, deep carving turns, snow particles flying into the air, sunlight flickering between trees, convincing speed and terrain response.

6    A motorcycle rider weaves through dense traffic in a narrow alley, mirrors narrowly missing obstacles, natural head turns, shaky chase camera, street vendors and pedestrians reacting believably.

7    A 6-second cinematic street portrait video, 24fps: woman in a vibrant market, saturated fruit stalls (reds, oranges, greens) behind; slow tracking shot as she walks, she glances to camera; bright daylight, high dynamic range, crisp detail, shallow depth of field

8    A 6-second cinematic portrait video, 24fps, golden hour: elderly man with warm smile, colorful scarf fluttering lightly; slow push-in, shallow depth of field, sun flare at frame edge, saturated oranges and teals, detailed wrinkles and skin texture, soft background bokeh

9    A 6-second animal video, 24fps: samoyed wearing a bright red bandana in a sunlit forest clearing; sunbeams through leaves, camera slowly pushes in as the dog tilts head; saturated greens, warm highlights, detailed fluffy fur

10   A 6-second cinematic close-up, 24fps: young man with freckles and bright green eyes under clear sunlight; gentle rack focus from lips to eyes, saturated teal background with warm rim light, ultra sharp iris detail, natural skin texture

## C. Limitations and future directions

**Detailed discussion on cache-aware schedule alignment scope.** As noted in the main paper, cache-aware schedule alignment is currently validated on image diffusion models. Here we elaborate on the computational reason and our preliminary attempts on video models. The alignment stage relies on outcome-driven distillation that backpropagates through the entire cached sampling trajectory to compute gradients with respect to the learnable time steps $\sigma$. For high-resolution video diffusion models, this end-to-end differentiation incurs prohibitive peak memory and wall-clock cost. We conducted preliminary experiments on Wan2.1-T2V-1.3B (Wan et al., 2025) with a limited calibration budget of 100 prompts and did not observe measurable gains; this may be related to insufficient calibration scale and the higher-dimensional trajectory space of video generation.

**Scaling the calibration set.** Calibration scale is a natural axis to study for both image and video diffusion. On image models such as FLUX.1-dev (Labs, 2024), where 100 prompts already produce consistent gains, larger and more diverse calibration data may further close the gap to the full-precision teacher and improve robustness at even tighter budgets. For video diffusion, the higher-dimensional trajectory space likely demands substantially more samples for a stable alignment signal. We plan to investigate both regimes with scaled-up calibration once compute permits, which would also help disentangle the contribution of calibration scale from factors intrinsic to the video domain.

**Understanding the cacheability of few-step models.** A more fundamental open question is why step-level caching remains effective on already-accelerated few-step models such as Hyper-SD (Ren et al., 2024). These models compress the full denoising trajectory into 4 to 8 steps via distillation, yet our cache policy continues to extract meaningful additional speedups on top of this acceleration (see Figure 6 and Table 6). This suggests that the temporal redundancy exploited by caching is partly orthogonal to the redundancy removed by few-step distillation. Characterizing what residual structure remains in the compressed trajectory, and how it interacts with the distillation objective, would shed light on the achievable limits of training-free acceleration.

*Table 8.* **Efficiency of Global Strategy Search.** Wall-clock time required to identify the caching policy $\mathbf{m}^*$ under different configurations. Here, $R$ denotes the number of independent search restarts, and $SA$ denotes the number of iterations in the Simulated Annealing phase. All experiments are conducted on a single NVIDIA H100 GPU.

| Config | Budget ($B$) | Evaluations | Search Time |
|---|---|---|---|
| R1 SA20 | 8 NFE | 94 | 5m 57s |
| R1 SA20 | 9 NFE | 125 | 8m 55s |
| R1 SA20 | 10 NFE | 127 | 10m 04s |
| R2 SA100 | 8 NFE | 290 | 18m 23s |
| R2 SA100 | 9 NFE | 365 | 26m 01s |
| R2 SA100 | 10 NFE | 427 | 33m 47s |

*Table 9.* **Impact of Search Intensity on Generation Quality.** We compare the lightweight search (approx. 10 mins) against the standard intensive search (approx. 30 mins) under a 9-NFE budget. The performance gap is negligible, demonstrating the efficiency and robustness of our hybrid optimization.

| Search Intensity | Time | LPIPS ↓ | PSNR ↑ | SSIM ↑ | ImageReward ↑ |
|---|---|---|---|---|---|
| Lightweight (R1 SA20) | ~10m | 0.2020 | 23.586 | 0.8201 | 0.9633 |
| Standard (R2 SA100) | ~30m | 0.2036 | 23.722 | 0.8290 | 0.9737 |
| **Relative Gap** | - | -0.81% | 0.58% | 1.08% | 1.07% |

*Table 10.* **Efficiency of Offline Schedule Calibration.** The wall-clock time required to optimize the time schedule for different NFE budgets. All calibration runs are conducted on 4×NVIDIA H100 GPUs using only 100 prompts for 1 epoch.

| Target Budget | Device | Calibration Time |
|---|---|---|
| 5 NFE | 4×H100 | 3m 46s |
| 6 NFE | 4×H100 | 4m 27s |
| 7 NFE | 4×H100 | 5m 38s |

*Table 11.* **Search-Transfer Generalization across Inference Settings**. A cache configuration searched in one source setting is directly transferred to other solvers, resolutions, and CFG scales on FLUX.1-dev (Labs, 2024) using DrawBench (Saharia et al., 2022) prompts, demonstrating consistent generalization of the searched configuration across inference settings.

| Setting | Method | Efficiency | Visual Quality | | | | | |
|---|---|---|---|---|---|---|---|---|
| | | Latency (s)↓ | LPIPS↓ | SSIM↑ | PSNR↑ | ImageReward↑ | CLIP(L)↑ | HPSv2.1↑ |
| iPNDM(2M) | Original: 28 steps | 6.90 | – | – | – | 0.9740 | 27.48 | 0.2984 |
| | TeaCache | 2.75 | 0.4124 | 0.6569 | 15.35 | 0.9387 | 27.52 | 0.2908 |
| | `BudCache-base` | 2.75 | **0.1632** | **0.8337** | **23.80** | **0.9484** | **27.57** | **0.2959** |
| DPM-Solver++(2M) | Original: 28 steps | 6.90 | – | – | – | 0.9857 | 27.35 | 0.2982 |
| | TeaCache | 2.75 | 0.3912 | 0.6717 | 15.97 | 0.9576 | **27.46** | 0.2917 |
| | `BudCache-base` | 2.75 | **0.1668** | **0.8333** | **23.80** | **0.9778** | 27.40 | **0.2955** |
| 512×512 | Original: 28 steps | 6.90 | – | – | – | 0.9922 | 27.73 | 0.2972 |
| | TeaCache | 2.75 | 0.3154 | 0.6693 | 16.77 | 0.9289 | 27.29 | 0.2898 |
| | `BudCache-base` | 2.75 | **0.1167** | **0.8465** | **24.70** | **0.9787** | **27.71** | **0.2976** |
| 768×1024 | Original: 28 steps | 6.90 | – | – | – | 0.8817 | 27.40 | 0.2871 |
| | TeaCache | 2.75 | 0.3634 | 0.6969 | 16.91 | 0.8207 | 27.45 | 0.2772 |
| | `BudCache-base` | 2.75 | **0.1370** | **0.8663** | **25.47** | **0.8586** | **27.55** | **0.2859** |
| CFG = 5 | Original: 28 steps | 6.90 | – | – | – | 0.9776 | 27.37 | 0.2974 |
| | TeaCache | 2.75 | 0.3749 | 0.6878 | 16.49 | 0.9419 | **27.42** | 0.2898 |
| | `BudCache-base` | 2.75 | **0.1536** | **0.8560** | **24.75** | **0.9660** | 27.41 | **0.2949** |
| CFG = 7 | Original: 28 steps | 6.90 | – | – | – | 0.9642 | 27.37 | 0.3001 |
| | TeaCache | 2.75 | 0.4254 | 0.6656 | 15.60 | 0.8343 | 27.14 | 0.2838 |
| | `BudCache-base` | 2.75 | **0.1649** | **0.8380** | **24.42** | **0.9458** | **27.29** | **0.2957** |

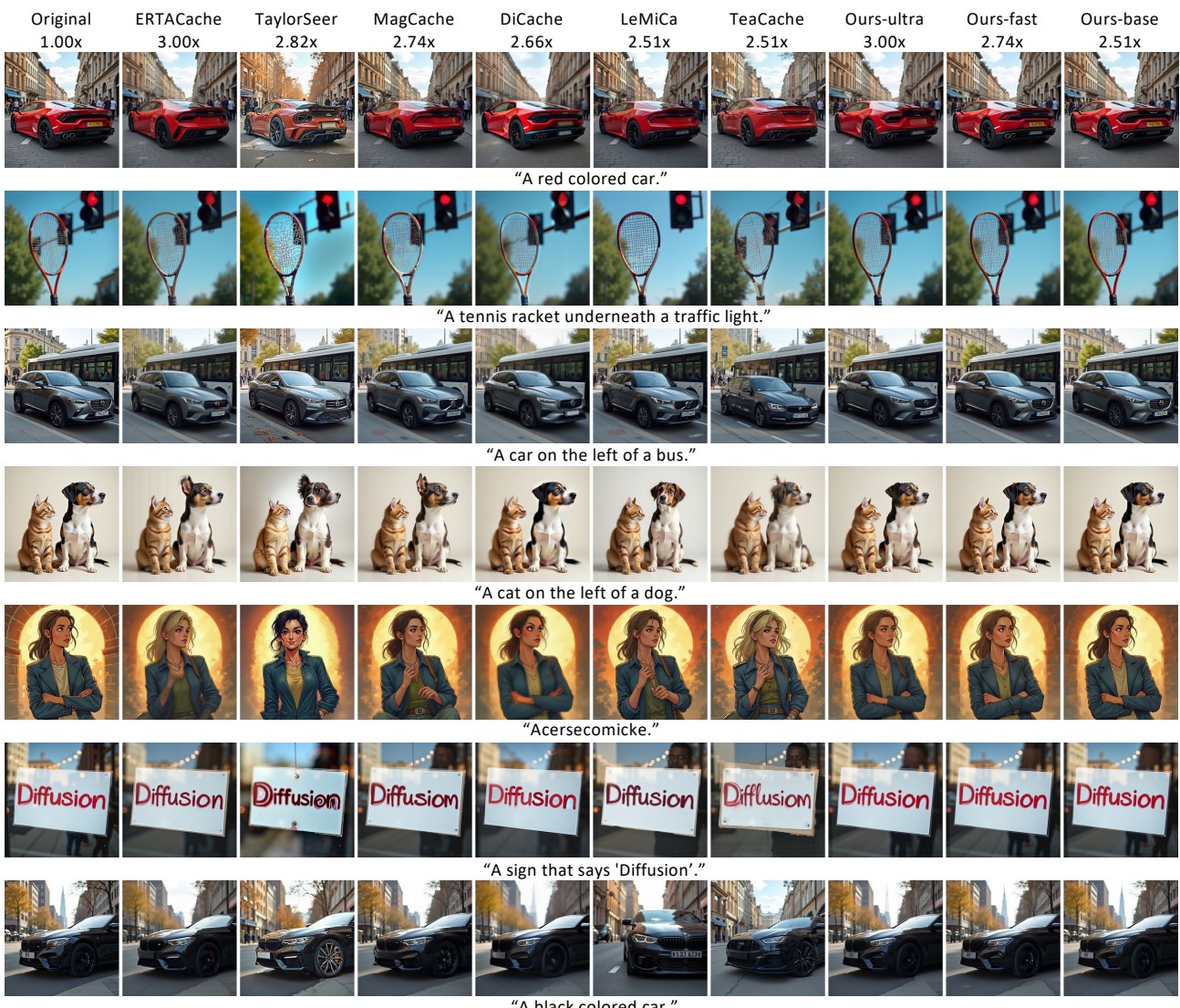

*Figure 8.* **Qualitative comparison with baselines on FLUX.1-dev** (Labs, 2024). BudCache better preserves semantic consistency and fine details, especially for challenging cases such as text rendering and complex geometric structures. Best viewed zoomed in.

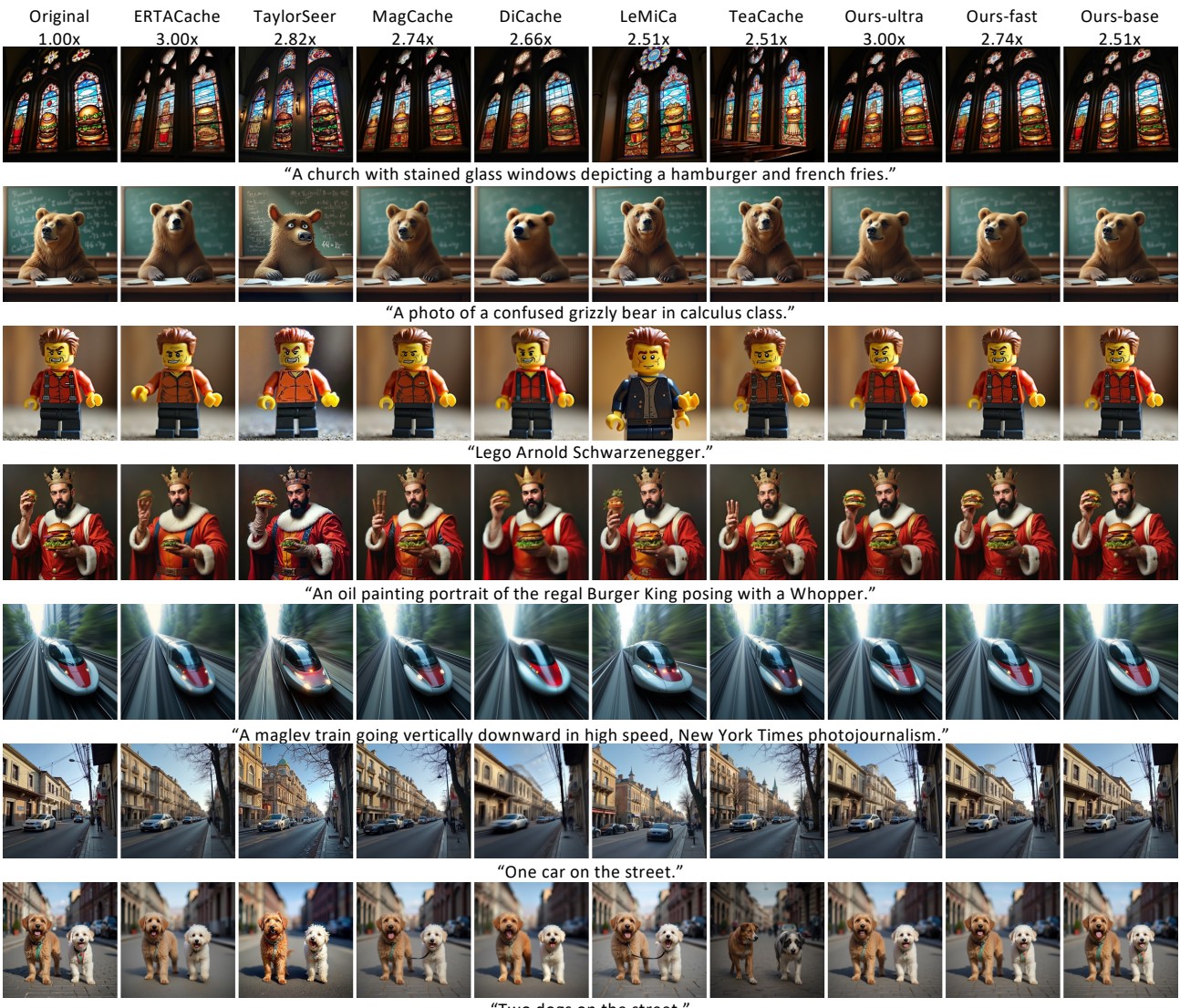

*Figure 9.* **Qualitative comparison with baselines on FLUX.1-dev** (Labs, 2024). `BudCache` better preserves semantic consistency and fine details, especially for challenging cases such as text rendering and complex geometric structures. Best viewed zoomed in.

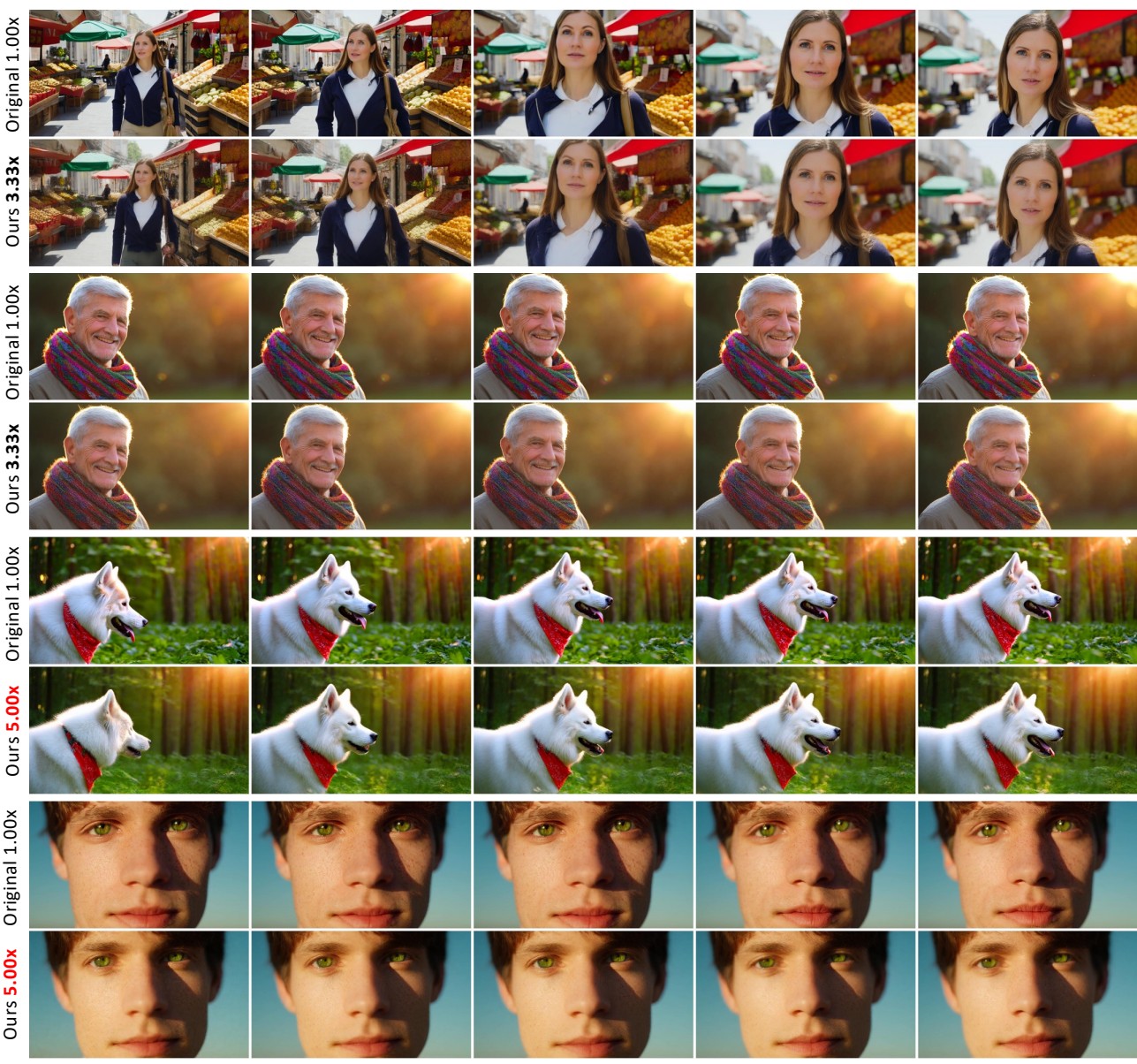

*Figure 10.* **Qualitative results on Wan2.1-T2V-1.3B** (Wan et al., 2025). `BudCache` achieves visually strong results even under a $5\times$ acceleration setting with only 10 NFE. It preserves semantic consistency and fine-grained visual details across frames, demonstrating effective fidelity preservation for video generation. Best viewed zoomed in.

