# OpenReview forum: "Budget-Constrained Step-Level Diffusion Caching"
_ICML.cc/2026/Conference — ICML 2026 regular_

### Official Review · Reviewer_WFRA · 2026-03-07

**Soundness:** 3
**Presentation:** 2
**Significance:** 4
**Originality:** 4
**Overall Recommendation:** 5
**Confidence:** 3

**Summary:**

This paper introduces BudCache, a budget-constrained optimization framework for step-level caching in diffusion models. The authors identify that existing caching strategies predominantly rely on reactive, myopic heuristic thresholds, which lead to unpredictable inference latency and suboptimal generation quality. To address this, BudCache formulates cache step selection as a global planning problem under a strict Number of Function Evaluations (NFE) budget. It employs a hybrid search strategy using Simulated Annealing for global exploration and Hill Climbing for deterministic refinement to find the optimal binary caching mask. Furthermore, the paper introduces an optional cache-aware schedule alignment phase using teacher-student distillation to correct trajectory drift caused by aggressive caching. Extensive experiments on FLUX.1-dev and Wan2.1 demonstrate that BudCache consistently outperforms state-of-the-art baselines (such as TeaCache and ERTACache) under identical latency constraints.

**Compliance With Llm Reviewing Policy:**

Affirmed.

**Final Justification:**

My concerns have been partially resolved, and authors promised to make corresponding modifications, so I will maintain my opinion of accept.

**Key Questions For Authors:**

1. Generalization of the Static Mask: The optimal caching mask $m^*$ is computed offline using a calibration set of 100 MS-COCO prompts. How sensitive is the globally optimized mask to vastly out-of-domain prompts (e.g., specialized medical domain, highly abstract art, or prompts requiring intense compositional reasoning)? Does a static mask struggle when the temporal redundancy of a specific input strongly deviates from the calibration distribution?
2. Performance on Video Models without Schedule Optimization: Given that the schedule refinement is computationally prohibitive for Wan2.1 (Appendix D) , can you explicitly clarify if the qualitative results in Figures 9 and 10 rely purely on the global strategy search (SA+HC) without schedule optimization? How much of a performance margin does BudCache maintain over baselines in the video domain when relying solely on the optimized mask?
3. Title Correction: Please correct the typo in the paper's main title from "Step-Leve" to "Step-Level". Really obvious typo.

**Limitations:**

yes

**Strengths And Weaknesses:**

Strengths: The mathematical formulation of step-level caching as a budget-constrained binary optimization problem is highly rigorous and well-motivated. The hybrid search mechanism (Simulated Annealing + Hill Climbing) effectively addresses the non-convex combinatorial search space, as empirically validated by the convergence analysis in Figure 4. The theoretical justification distinguishing step-level caching from standard skip-step solvers (Appendix A) provides excellent intuition regarding the "guidance" effect of the final layer.

Weaknesses: As acknowledged in Appendix D, the time schedule optimization relies on an end-to-end distillation process that is computationally prohibitive for video generation models (e.g., Wan2.1). Thus, the schedule alignment mechanism, while highly effective for image models, lacks comprehensive empirical validation in the video domain due to hardware constraints.

Originality: While Simulated Annealing, Hill Climbing, and distillation are well-known techniques, the synthesis of these methods to invert the caching paradigm—from reactive error-thresholding to global budget-constrained planning—is highly original. The integration of a schedule optimization objective specifically designed to counteract cache-induced trajectory drift provides a novel perspective on solver discretization.

---

> ### Author Rebuttal · Authors · 2026-03-31
>
> We thank the reviewer for the thoughtful feedback and constructive questions. We address your technical questions and the typo below:
>
> 1. Generalization of the static mask to strongly shifted prompt distributions.
>
> The 100 MS-COCO prompts mentioned in the appendix are used for the optional schedule optimization stage, rather than for the offline global mask search itself. For the search stage, we use the same calibration prompt setup as MagCache. We evaluate the same searched static mask, without re-optimization, on additional benchmarks and harder prompt distributions. Unless otherwise noted, the image-generation results below use the same 28-step FLUX setting and the **same acceleration ratio** as TeaCache. We first test GenEval. The results are shown below:
>
> |Setting|PSNR↑|SSIM↑|LPIPS↓|HPSv2.1↑|ImageReward↑|ClipScore↑|
> |:-:|:-:|:-:|:-:|:-:|:-:|:-:|
> |Full|-|-|-|0.3086|1.0046|27.84|
> |TeaCache|18.56|0.7586|0.3349|0.2995|0.9393|27.85|
> |Ours|**26.57**|**0.8802**|**0.1388**|**0.3055**|**0.9710**|27.83|
>
> We further test the same mask on 100 strictly OOD prompts (including specialized medical prompts and highly abstract prompts) and a separate set of 100 highly complex prompts (e.g., prompts requiring demanding compositional reasoning).
>
> OOD prompts:
> |Setting|PSNR↑|SSIM↑|LPIPS↓|
> |:-:|:-:|:-:|:-:|
> |Full|-|-|-|
> |TeaCache|18.37|0.7106|0.3559|
> |Ours|**23.68**|**0.8144**|**0.2105**|
>
> Highly complex prompts:
> |Setting|PSNR↑|SSIM↑|LPIPS↓|
> |:-:|:-:|:-:|:-:|
> |Full|-|-|-|
> |TeaCache|19.08|0.7392|0.3322|
> |Ours|**26.23**|**0.8670**|**0.1575**|
>
> Overall, these results suggest that the searched global mask does not simply overfit to the calibration prompt distribution, but remains effective under substantial prompt-distribution shift.
>
> In our response to Reviewer RQJA, we also provided extensive evidence showing this mask transfers seamlessly across different models (Wan-14B), solvers, and CFG scales.
>
> 2. Video performance without schedule optimization.
>
> To explicitly clarify: yes, the qualitative results in Figures 9 and 10 rely purely on the globally searched mask obtained via SA+HC, without any schedule optimization. To demonstrate the performance margin BudCache maintains over baselines in the video domain using just the mask, we provide the quantitative results below (evaluated on 100 prompts, with latency measured on an A800 GPU):
>
> |Setting|Latency↓|PSNR↑|SSIM↑|LPIPS↓|
> |:-:|:-:|:-:|:-:|:-:|
> |Full|189s|-|-|-|
> |TeaCache|100s|21.52|0.7494|0.1929|
> |Ours|**82s**|**25.93**|**0.8502**|**0.1167**|
>
> These results show that BudCache maintains a clear advantage over TeaCache in both latency and quality in the video domain.
>
> 3. Title Correction
>
> Thank you for catching this! We completely agree it is an obvious oversight. We will absolutely correct the title to "Step-Level" in the final revision.

---

> > ### Author Rebuttal · Reviewer_WFRA · 2026-04-03
> >
> > Thanks for authors' reply. My concerns have been partially resolved, so I will maintain my score.

---

### Official Review · Reviewer_RQJA · 2026-03-10

**Soundness:** 3
**Presentation:** 3
**Significance:** 3
**Originality:** 3
**Overall Recommendation:** 5
**Confidence:** 3

**Summary:**

This paper proposes a Budget-Constrained Step-Level Caching method for accelerating diffusion model inference. The authors reformulate the traditional cache decision process, which relies on online heuristic rules, as an offline global planning problem under a fixed computational budget: given an available NFE budget, they pre-search which denoising steps should be recomputed and which steps should reuse the cache, so as to balance speed and generation quality. The authors design an offline policy optimization pipeline combining SA and HC, and simultaneously perform Cache-Aware Schedule Alignment. Experiments show that the method can achieve better generation quality on both image and video generation tasks under the same or lower computational cost.

**Compliance With Llm Reviewing Policy:**

Affirmed.

**Key Questions For Authors:**

- Can the authors analyze, either theoretically or experimentally, whether the fixed caching policy obtained by offline search remains stable and effective across different prompts, seeds, guidance scales, or resolutions?
- In the comparison with existing heuristic methods, BudCache introduces additional offline search and schedule optimization. If similar offline tuning budget were given to the baselines, would the conclusion still hold?
- Since the method relies on matching the full-compute teacher trajectory, whether this objective is sufficiently reliable with respect to final perceptual quality, text alignment, and robustness?

**Limitations:**

yes

**Strengths And Weaknesses:**

Strength:
- Reformulating step-level caching as a budget-constrained global planning problem is an interesting perspective.
- The method design is complete. It not only optimizes which steps should be recomputed, but also further considers time-step schedule optimization coordinated with the caching mechanism, making the overall framework relatively systematic.
- This method has strong engineering value: it can provide more controllable inference budget and latency, which is more suitable for practical deployment than heuristic methods relying on online threshold triggering.

Weakness:
- The method assumes that a fixed offline caching plan can generalize well across different prompts and sampling trajectories, but this is a relatively strong assumption and is not sufficiently validated, although I am tend to believe this is a good assumption.
- Part of the performance gain may come from the additional offline search and tuning cost, so the comparison with heuristic baselines may not be fully fair.
- The scalability is not entirely clear. For example, if the method is applied to different samplers, different resolutions, or larger models, would the caching policy need to be re-optimized?

Presentation Weakness (personal suggestion, does not affect the score):
- Figure 2 looks quite confusing. Although one can understand what the figure is trying to express after reading the whole paper, I hope the authors can provide a more concise framework figure that can more precisely present the core contribution. Stacking formulas in the figure is not a good choice.

---

> ### Author Rebuttal · Authors · 2026-03-31
>
> We thank the reviewer for the thoughtful feedback and recognizing the engineering value of our approach. We address your specific concerns regarding generalization, fairness, and scalability below:
>
> 1. Robustness of the fixed offline-searched policy across prompts.
>
> You rightly pointed out that assuming a fixed offline policy generalizes well is a strong assumption. To empirically validate this, we took the exact same searched static mask, without any re-optimization, and evaluated it on entirely different benchmarks and much harder, Out-of-Distribution (OOD) prompt distributions, all at the same acceleration ratio as TeaCache.
>
> GenEval benchmark:
> |Setting|PSNR↑|SSIM↑|LPIPS↓|HPSv2.1↑|ImageReward↑|ClipScore↑|
> |:-:|:-:|:-:|:-:|:-:|:-:|:-:|
> |Full|-|-|-|0.3086|1.0046|27.84|
> |TeaCache|18.56|0.7586|0.3349|0.2995|0.9393|27.85|
> |Ours|26.57|0.8802|0.1388|0.3055|0.9710|27.83|
>
> We then tested 100 severe OOD prompts and 100 highly complex compositional prompts. In all cases, the static mask exhibited remarkable stability and outperformed the baseline significantly:
>
> OOD Prompts:
> |Setting|PSNR↑|SSIM↑|LPIPS↓|
> |:-:|:-:|:-:|:-:|
> |Full|-|-|-|
> |TeaCache|18.37|0.7106|0.3559|
> |Ours|23.68|0.8144|0.2105|
>
> Highly complex prompts:
> |Setting|PSNR↑|SSIM↑|LPIPS↓|
> |:-:|:-:|:-:|:-:|
> |Full|-|-|-|
> |TeaCache|19.08|0.7392|0.3322|
> |Ours|26.23|0.8670|0.1575|
>
> 2. Scalability across models / solvers / sampling conditions.
>
> This is a critical question for practical deployment. To test scalability across model sizes, we applied a mask searched on Wan-1.3B directly to the much larger Wan-14B model. Evaluated on a curated set of 30 prompts, the transferred mask yielded both faster inference and better quantitative performance than TeaCache:
>
> |Setting|Latency↓|PSNR↑|SSIM↑|LPIPS↓|
> |:-:|:-:|:-:|:-:|:-:|
> |Full|515s|-|-|-|
> |TeaCache|291s|26.70|0.8932|0.0639|
> |Ours|223s|27.94|0.8932|0.0618|
>
> Furthermore, to test whether the searched mask remains effective across varying generation conditions (different solvers, resolutions, and CFG scales), we evaluated the same mask under a wide variety of FLUX settings. BudCache consistently outperforms the baseline in all cases (Full: 28-step; same acceleration ratio as TeaCache).
>
> Cross-solver transfer:
> |Solver|Method|PSNR↑|SSIM↑|LPIPS↓|HPSv2.1↑|ImageReward↑|ClipScore↑|
> |:-:|:-:|:-:|:-:|:-:|:-:|:-:|:-:|
> |iPNDM(2M)|Full|-|-|-|0.2984|0.9740|27.48|
> ||TeaCache|15.35|0.6569|0.4124|0.2908|0.9387|27.52|
> ||Ours|23.80|0.8337|0.1632|0.2959|0.9484|27.57|
> |DPM-Solver++(2M)|Full|-|-|-|0.2982|0.9857|27.35|
> ||TeaCache|15.97|0.6717|0.3912|0.2917|0.9576|27.46|
> ||Ours|23.80|0.8333|0.1668|0.2955|0.9778|27.40|
>
> Cross-resolution transfer:
> |Resolution|Method|PSNR↑|SSIM↑|LPIPS↓|HPSv2.1↑|ImageReward↑|ClipScore↑|
> |:-:|:-:|:-:|:-:|:-:|:-:|:-:|:-:|
> |512x512|Full|-|-|-|0.2972|0.9922|27.73|
> ||TeaCache|16.77|0.6693|0.3154|0.2898|0.9289|27.29|
> ||Ours|24.70|0.8465|0.1167|0.2976|0.9787|27.71|
> |768x1024|Full|-|-|-|0.2871|0.8817|27.40|
> ||TeaCache|16.91|0.6969|0.3634|0.2772|0.8207|27.45|
> ||Ours|25.47|0.8663|0.1370|0.2859|0.8586|27.55|
>
> Different guidance scales:
> |CFG scale|Method|PSNR↑|SSIM↑|LPIPS↓|HPSv2.1↑|ImageReward↑|ClipScore↑|
> |:-:|:-:|:-:|:-:|:-:|:-:|:-:|:-:|
> |CFG=5|Full|-|-|-|0.2974|0.9776|27.37|
> ||TeaCache|16.49|0.6878|0.3749|0.2898|0.9419|27.42|
> ||Ours|24.75|0.8560|0.1536|0.2949|0.9660|27.41|
> |CFG=7|Full|-|-|-|0.3001|0.9642|27.37|
> ||TeaCache|15.60|0.6656|0.4254|0.2838|0.8343|27.14|
> ||Ours|24.42|0.8380|0.1649|0.2957|0.9458|27.29|
>
> 3. Fairness with respect to offline cost and tuning budget.
>
> It is important to clarify that our method does not train the model weights; the offline calibration is extremely lightweight and done only once. Moreover, it is worth noting that baseline methods are not entirely "cost-free" prior to deployment. For example, to ensure stable dynamic thresholding, TeaCache requires processing 70 prompts to obtain a robust polynomial fit. Because our offline search is similarly cheap but yields a globally optimal, reusable mask that completely avoids online latency, we believe the quality-speed trade-off strongly justifies our framework over purely heuristic baselines.
>
> 4. Reliability of the teacher-trajectory objective.
>
> We agree that forcing a model to simply mimic a teacher could be risky if we only looked at basic pixel reconstruction. However, we do not rely solely on reconstruction metrics like PSNR. We rigorously evaluate using ImageReward, CLIPScore, and HPSv2.1 to ensure semantic alignment and human preference are maintained. Furthermore, Table 2 in our main text demonstrates that combining the mask with schedule optimization (BudCache-Opt) consistently outperforms the mask alone (BudCache) at 5, 6, and 7 NFE across all perceptual metrics. This shows our objective successfully translates into superior visual quality, rather than just overfitting to a proxy loss.
>
> Thank you also for the suggestion on Figure 2. We agree that the current figure can be simplified, and we will revise it accordingly.

---

> > ### Author Rebuttal · Reviewer_RQJA · 2026-04-03
> >
> > The authors have responded and clarified raised questions.

---

### Official Review · Reviewer_snXE · 2026-03-12

**Soundness:** 3
**Presentation:** 3
**Significance:** 3
**Originality:** 3
**Overall Recommendation:** 4
**Confidence:** 3

**Summary:**

This paper proposes BudCache, a method to speed up diffusion image generation by carefully deciding which denoising steps actually need to run the neural network and which ones can reuse cached results. Instead of using heuristic rules that trigger recomputation based on local errors (which can lead to unpredictable latency and suboptimal quality), the authors fix a strict computation budget and treat the problem as a global optimization task: choose the most important steps to compute so that the final image quality is maximized. They search for this optimal step pattern offline using a combination of simulated annealing and hill climbing, producing a deterministic caching policy that adds no overhead during inference. To further improve quality when many steps are skipped, they also adjust the diffusion time schedule using a teacher–student distillation objective so the cached solver better follows the original trajectory. Experiments on modern diffusion models show that this approach achieves around 2.5–3× speedup while preserving image quality.

**Compliance With Llm Reviewing Policy:**

Affirmed.

**Final Justification:**

My concerns have been addressed, I will maintain my score.

**Key Questions For Authors:**

1. How did you train or calibrate the caching policy during the offline optimization stage? Specifically, which dataset or prompt distribution was used for the search process, and approximately how many prompts were required to obtain a stable caching policy?
2. The proposed method learns a single static caching policy through offline optimization and assumes that it generalizes across different prompts. How sensitive is the learned policy to the prompt distribution used during calibration? Have you evaluated the policy on prompts from distributions that are significantly different from the calibration set?
3. Have you considered hybrid approaches that combine global optimization with lightweight runtime adaptation? Insights into this design tradeoff could further strengthen the method.

**Limitations:**

yes

**Strengths And Weaknesses:**

## Strengths

- The method design is technically sound and practical, using a hybrid search strategy (Simulated Annealing + Hill Climbing) to efficiently explore the combinatorial space of caching policies.
- Clearly formulates step-level diffusion caching as a budget-constrained optimization problem, addressing limitations of prior heuristic methods that rely on local thresholds and unpredictable recomputation.
- The approach is training-free and compatible with pretrained diffusion models, making it easy to deploy without retraining or modifying the model architecture. The optimization is performed offline, ensuring that the method introduces no additional inference overhead during deployment.
- The paper provides a clear motivation and reframes diffusion acceleration as a resource allocation problem, offering a useful perspective for optimizing inference efficiency.

## Weaknesses


- The evaluation uses a limited set of test datasets, which raises concerns about whether the learned caching policy generalizes well to different prompt distributions, datasets, or diffusion models.
- The speedup improvement over prior work is relatively modest. Although the method shows consistent gains, the performance improvement compared to existing caching methods may not be substantial enough to clearly justify the added optimization procedure.
- The overall novelty is moderate, as most components (e.g., simulated annealing, hill climbing, and schedule optimization via distillation) are existing techniques. The main contribution lies in formulating diffusion caching as a global search problem and combining these techniques for diffusion caching.
- The method formulates diffusion caching as a global search problem and assumes that a single static caching policy can generalize across different prompts. This assumption may be questionable in practice. A coarse-grained policy could lead to over-caching in some cases, sacrificing image quality, or under-caching in others, limiting performance gains. In contrast, prior work uses runtime thresholds to dynamically decide whether to reuse cached results. While this introduces some runtime overhead, it provides more adaptive behavior across different inputs.
- The approach requires an offline search/training stage to determine the caching policy, and this process must be repeated for different models and potentially for different total numbers of denoising steps. This reduces the flexibility of the method and may introduce additional engineering overhead when deploying the approach across multiple diffusion models or sampling configurations.

---

> ### Author Rebuttal · Authors · 2026-03-31
>
> We thank the reviewer for the thoughtful comments and constructive questions. We have grouped your main concerns into four areas and address them below:
>
> 1. Calibration setup and prompt efficiency.
>
> Regarding the calibration stage, we used the same calibration prompt setup as MagCache to ensure a fair baseline comparison. The results reported in the paper are obtained by searching with a single calibration prompt, which already yields a stable cache mask in our setting. We further verified that this searched mask generalizes well across multiple datasets and different generation settings. In addition, in the few-step Hyper-SD setting, two different prompts led to the exact same cache mask (selecting steps 2, 4, 5, and 7), further supporting the stability of the offline search.
>
> 2. Generalization of a static mask.
>
> This is a very valid concern. To verify that our static mask does not overfit, we evaluated the exact same searched mask, without any re-optimization, on completely different benchmarks and much harder prompt distributions, while **keeping the acceleration ratio identical to TeaCache**.
>
> First, we tested on the GenEval benchmark, where our method outperforms the baseline:
>
> |Setting|PSNR↑|SSIM↑|LPIPS↓|HPSv2.1↑|ImageReward↑|ClipScore↑|
> |:-:|:-:|:-:|:-:|:-:|:-:|:-:|
> |Full|-|-|-|0.3086|1.0046|27.84|
> |TeaCache|18.56|0.7586|0.3349|0.2995|0.9393|27.85|
> |Ours|**26.57**|**0.8802**|**0.1388**|**0.3055**|**0.9710**|27.83|
>
> Next, we pushed the limits by testing the mask on 100 Out-of-Distribution (OOD) prompts, and subsequently on a set of 100 highly complex prompts. In both extreme scenarios, BudCache consistently outperformed the baseline in reconstruction quality:
>
> OOD prompts:
> |Setting|PSNR↑|SSIM↑|LPIPS↓|
> |:-:|:-:|:-:|:-:|
> |Full|-|-|-|
> |TeaCache|18.37|0.7106|0.3559|
> |Ours|**23.68**|**0.8144**|**0.2105**|
>
> Highly complex prompts:
> |Setting|PSNR↑|SSIM↑|LPIPS↓|
> |:-:|:-:|:-:|:-:|
> |Full|-|-|-|
> |TeaCache|19.08|0.7392|0.3322|
> |Ours|**26.23**|**0.8670**|**0.1575**|
>
> These results suggest that the searched mask does not overfit to the calibration prompt distribution and remains effective under substantial prompt-distribution shift.
>
> For these deliberately out-of-distribution prompts, we do not report image-quality metrics, since the reward models themselves may also undergo domain shift in this setting. We therefore focus on reference-based reconstruction metrics here, which remain directly grounded by the full-compute output.
>
> 3. Practical value vs. prior dynamic methods.
>
> Building on the previous point, it is worth noting that the calibration in BudCache is a one-time, lightweight process (as detailed in our Appendix). Furthermore, many prior baseline methods also require non-trivial offline setups; for instance, TeaCache requires 70 prompts to obtain a robust polynomial fit. Therefore, the practical value of BudCache lies in the fact that its calibration cost is low, the resulting policy is highly reusable (as supported by the OOD experiments above), and it completely removes unpredictable online decision latency during deployment.
>
> 4. Hybrid designs.
>
> We find your suggestion of a hybrid design very insightful. This touches on a fundamental trade-off between global policies (like ours) and instance-level policies (like TeaCache). BudCache is intentionally built around a globally optimized mask to guarantee zero decision overhead at runtime. Online adaptation, while flexible, introduces runtime costs and can be hypersensitive to threshold tuning. That being said, we completely agree that blending a globally optimized baseline mask with a lightweight, instance-level adaptation module is a highly promising direction for balancing optimality, adaptivity, and efficiency in future work.

---

> > ### Author Rebuttal · Reviewer_snXE · 2026-04-03
> >
> > Thanks for the response, my concerns have been addressed.

---

### Official Review · Reviewer_nkWx · 2026-03-19

**Soundness:** 3
**Presentation:** 3
**Significance:** 2
**Originality:** 3
**Overall Recommendation:** 4
**Confidence:** 3

**Summary:**

- This paper introduces BudCache, which considers not only an error threshold when caching diffusion steps, but also computational cost and final image quality. The authors employ simulated annealing and hill-climbing methods to search for a globally optimal caching policy, and further refine the approach by optimizing time discretization via distillation to compensate for aggressive caching. The method is evaluated across a range of experimental settings.

**Compliance With Llm Reviewing Policy:**

Affirmed.

**Final Justification:**

The authors sufficiently addressed my concerns, and I thus maintain my score.

**Key Questions For Authors:**

Please refer to the weaknesses.

**Limitations:**

Yes.

**Strengths And Weaknesses:**

Strengths
- The paper addresses a timely and relevant problem and is supported by solid empirical evaluations.
- The writing is clear, well organized, and easy to follow.
- Experiments include not only simple image domain but also video dataset.

Weaknesses
- A primary concern is whether the comparison with baselines is conducted fairly. For instance, in Table 1 and Figure 3, it is unclear whether the baseline caching methods involve distillation or additional fine-tuning. It would be helpful if the authors could also report the training budget alongside these results.
- It is unclear whether BudCache can be combined with other efficiency-oriented techniques for diffusion models, such as progressive distillation or consistency models, particularly in the low-NFE regime.
- Regarding the impact of the search space size, the observed inverted U-shaped trend is not well explained. Additional intuition or analysis would help clarify why this behavior is expected.
- It is unclear why FID is not reported, given that it is a standard evaluation metric in diffusion-based image generation.
- It would be useful to clarify whether the findings in Appendix A.4 (guidance effect of the final layer) generalize across different diffusion architectures, such as U-Net.

---

> ### Author Rebuttal · Authors · 2026-03-31
>
> We sincerely thank the reviewer for the thoughtful comments and constructive questions. We have carefully considered your feedback and address your specific concerns below:
>
> 1. Fairness and training budget.
>
> To ensure a fair comparison, we clarify that the results reported in Table 1 and Figure 3 do not use distillation or additional fine-tuning, and only use the offline search proposed in our method. It is worth noting that our method does not require training the model itself; rather, the calibration is a one-time, lightweight search. We have detailed the full time cost for this in Appendix Tables 3 and 5. Furthermore, many baseline methods in our comparison also require setup-specific tuning or offline searches. For instance, TeaCache requires processing 70 prompts just to obtain a robust polynomial fitting. Our offline process is lightweight and the resulting mask remains robust across multiple datasets and Out-of-Distribution (OOD) settings.
>
> 2. Compatibility with low-NFE / distilled models.
>
> We agree that compatibility with other efficiency techniques is a great point. To address this, we evaluated BudCache on a representative distilled few-step model, Hyper-SD. Specifically, we used the 8-step setting and compared BudCache with TeaCache at the same acceleration ratio on DrawBench prompts. All latency results below are measured on a single H100 GPU. As shown below, BudCache maintains a clear advantage:
>
> |Setting|Latency↓|PSNR↑|SSIM↑|LPIPS↓|HPSv2.1↑|ImageReward↑|ClipScore↑|
> |:-:|:-:|:-:|:-:|:-:|:-:|:-:|:-:|
> |Full(8 steps)|2.07s|-|-|-|0.3076|1.0175|27.43|
> |TeaCache|1.28s|21.56|0.7631|0.1982|0.3038|0.9873|27.43|
> |Ours|1.28s|**25.28**|**0.8122**|**0.1417**|**0.3061**|**1.0096**|**27.58**|
>
> These results demonstrate that BudCache can be directly applied to distilled few-step models, while still providing better quality than TeaCache.
>
> 3. Why the search-space-size curve can be inverted-U shaped.
>
> You raised a great question regarding the trend in Figure 5. Intuitively, under a fixed computational budget, increasing the total logical steps $K$ initially reduces the solver's inherent discretization error, which naturally improves generation quality. However, as $K$ continues to grow, we are forced to cache and reuse features for a larger portion of the steps. This high effective caching ratio accumulates excessive trajectory drift and cache-induced errors. Ultimately, these errors outweigh the benefits of a finer solver discretization, causing the final quality to drop. This delicate balance creates the expected inverted-U shape.
>
> 4. FID.
>
> We primarily reported reference-based metrics because diffusion caching has a natural full-compute reference, and related caching works commonly focus on reconstruction/fidelity measures for this reason. We understand that FID remains a standard metric in the community, and we are happy to provide it. We evaluated FID $\downarrow$ on 5K MS-COCO images at the identical acceleration ratio. Under the **same acceleration ratio**, BudCache achieves a lower FID than TeaCache, which is consistent with our other quality metrics.
>
> ||Full|TeaCache|Ours|
> |:-:|:-:|:-:|:-:|
> |FID↓|37.4409|37.5358|**37.3668**|
>
> 5. Scope of Appendix A.4 and architectural generality.
>
> The purpose of Appendix A.4 is to explain the core difference between a 10-NFE solver and a 10-NFE cache method. This difference is closely related to the $\Delta$-DiT[1] discussion: it is fundamentally a discussion of **what should be cached**, and this strategy is particularly suitable for DiT-style backbones. We do not claim that the same guidance effect automatically generalizes to other diffusion architectures such as U-Net. At the same time, this scope is still practically meaningful, since several recent representative caching works, including TeaCache[2], MagCache[3], and LeMiCa[4], are also studied in this DiT-style setting. We will revise Appendix A.4 to make this scope more explicit.
>
> [1]$\Delta$-DiT: A Training-Free Acceleration Method Tailored for Diffusion Transformers
>
> [2] Timestep Embedding Tells: It's Time to Cache for Video Diffusion Model
>
> [3] MagCache: Fast Video Generation with Magnitude-Aware Cache
>
> [4] LeMiCa: Lexicographic Minimax Path Caching for Efficient Diffusion-Based Video Generation

---

> > ### Author Rebuttal · Reviewer_nkWx · 2026-04-03
> >
> > The authors sufficiently addressed my concerns, and I thus maintain my score.

---

### Decision · Program_Chairs · 2026-04-30

**Decision:**

Accept (regular)

**Comment:**

The paper received a clearly positive assessment overall. Reviewers found the core idea—reframing step-level diffusion caching as a budget-constrained global optimization problem, which is technically sound and practically valuable. The combination of offline SA+HC search with cache-aware schedule alignment was viewed as a coherent framework that improves controllability under strict latency/NFE budgets, with convincing gains on both image and video generation. Main weaknesses concern the fairness of baseline comparisons, limited validation of static-mask generalization across prompt distributions/models, missing FID and some scalability analysis, and the partial validation of schedule alignment on video. Still, reviewers agreed these issues do not outweigh the contribution.